# Mitofusin-2 stabilizes adherens junctions and suppresses endothelial inflammation via modulation of β-catenin signaling

Young-Mee Kim [1,2,3✉], Sarah Krantz[1,2], Ankit Jambusaria [1,2,4], Peter T. Toth [2,5], Hyung-Geun Moon[6], Isuru Gunarathna[1,4], Gye Young Park[6] & Jalees Rehman [1,2,3,4✉]

Endothelial barrier integrity is ensured by the stability of the adherens junction (AJ) complexes comprised of vascular endothelial (VE)-cadherin as well as accessory proteins such as β-catenin and p120-catenin. Disruption of the endothelial barrier due to disassembly of AJs results in tissue edema and the influx of inflammatory cells. Using three-dimensional structured illumination microscopy, we observe that the mitochondrial protein Mitofusin-2 (Mfn2) co-localizes at the plasma membrane with VE-cadherin and β-catenin in endothelial cells during homeostasis. Upon inflammatory stimulation, Mfn2 is sulfenylated, the Mfn2/β-catenin complex disassociates from the AJs and Mfn2 accumulates in the nucleus where Mfn2 negatively regulates the transcriptional activity of β-catenin. Endothelial-specific deletion of Mfn2 results in inflammatory activation, indicating an anti-inflammatory role of Mfn2 in vivo. Our results suggest that Mfn2 acts in a non-canonical manner to suppress the inflammatory response by stabilizing cell–cell adherens junctions and by binding to the transcriptional activator β-catenin.

[1] Division of Cardiology, Department of Medicine, University of Illinois at Chicago, Chicago, IL, USA. [2] Department of Pharmacology and Regenerative Medicine, University of Illinois at Chicago, Chicago, IL, USA. [3] University of Illinois Cancer Center, University of Illinois at Chicago, Chicago, IL, USA. [4] Department of Bioengineering, University of Illinois at Chicago, Chicago, IL, USA. [5] Research Resources Center, University of Illinois at Chicago, Chicago, IL, USA. [6] Division of Pulmonary, Critical Care, Sleep and Allergy, Department of Medicine, University of Illinois at Chicago, Chicago, IL, USA. ✉email: youngmee@uic.edu; jalees@uic.edu

One devastating manifestation of the disassembly of endothelial adherens junctions (AJs) is acute lung injury (ALI), in which an excessive immune response triggers the disruption of the lung endothelial barrier, fluid, and protein leak into the alveolar space resulting in compromised oxygen exchange and an unhinged state of inflammatory activation[1,2]. Understanding the mechanisms leading to the breakdown of endothelial barrier function as well as the inflammatory pathways within the endothelium are essential for developing therapeutic strategies to treat ALI patients[3]. The vascular endothelium maintains an intact barrier to prevent leakage of circulating nutrients, solutes, and fluid into the tissues as well as tightly regulating the influx of immune cells[4]. However, during severe infections and subsequent inflammatory responses, the endothelial barrier is compromised due to the breakdown of endothelial AJs. The plasma membrane AJ protein complex in endothelial cells consists of VE-cadherin as well as members of the catenin family such as β-catenin and p120-catenin[5]. The influx of immune cells following AJ disassembly exacerbates the inflammatory process, leading to a feed-forward activation of inflammation and further destruction of the endothelial barrier[1].

There are multiple mechanisms underlying the breakdown of AJs such as signaling induced by the pro-inflammatory cytokine TNFα which promotes inflammation via the generation of reactive oxygen species (ROS)[6,7]. ROS mainly mediate reversible or irreversible oxidative modification at cysteine residues[8]. Reactive cysteine thiol (SH) converts cysteine sulfenic acid (Cys-SOH), termed as protein sulfenylation, a key initial mediator of redox signaling[9]. Prior studies of inflammatory redox signaling and barrier function have understandably focused on known junctional proteins at the plasma membrane. However, recent studies on inflammatory signaling implicate the involvement of mitochondria[10–12].

The mitochondrial GTPases Mitofusin-1 (Mfn1) and Mitofusin-2 (Mfn2) are key regulators of mitochondrial function by mediating mitochondrial network fusion which allows for the distribution of proteins, mitochondrial DNA, and metabolites to maintain network connectivity[13,14]. Mitofusin 1 and Mitofusin 2 are located in the outer membrane of mitochondria and thus may facilitate interactions with other organelles[13]. The function of Mfn1/2 can be regulated by post-translational modifications such as ubiquitination, acetylation, or phosphorylation[13]. Mfn1/2 also acts as tethers for mitochondria with each other or with other organelles such as ER and as mitochondrial anchoring proteins[15]. Although Mfn1 and Mfn2 demonstrate approximately 80% sequence homology, double mutant embryos die earlier than either single mutant, and the post-natal Mfn1 knockout (KO) mice do not exhibit significant pathology whereas Mfn2 KO mice show rapid lethality[16]. It has thus been suggested that non-redundant functions of Mfn1 and Mfn2 that may go beyond the traditionally ascribed mitochondrial fusion roles of these proteins. Non-fusion unique roles of Mfn2 include the mediation of mitophagy and apoptosis[13], or regulation of contact sites with ER[14,16,17]. However, the potential roles of Mfn2 in the formation or stabilization of adherens junctions are not yet known.

In this study, we identified a non-canonical function of Mfn2 as a stabilizer of endothelial adherens junction complexes during homeostasis. Mfn2 is sulfenylated during inflammation and accumulates in the nucleus where it acts as an endogenous suppressor of inflammation by binding the transcriptional regulator β-catenin.

## Results

**Mfn2 binds to the adherens junction complex during homeostasis.** To identify binding partners of Mfn2 in an unbiased manner, we performed a comprehensive proteomic analysis in human lung microvascular endothelial cells (HLMVECs). Mfn2 was overexpressed in HLMVECs using lentiviral GFP-Mfn2 and GFP-Mfn2 was immunoprecipitated with GFP-trap magnetic agarose to reduce non-specific binding that may be present with the use of anti-GFP antibodies. Overexpression of lentiviral GFP in HLMVECs was used as a control to assess protein partners that non-specifically bind to GFP (Supplementary Fig. 1a, b). We focused on protein partners unique to the expression of GFP-Mfn2 versus the GFP-control. The obtained proteins were analyzed using high-affinity liquid chromatography with tandem mass spectrometry (LC-MS/MS). We considered candidates which are identified in at least two samples among three independent experiments with a >1.5-fold increase threshold. We found that 25 proteins specifically interacted with Mfn2 in ECs (see Source data) and all identified protein partners were classified based on their function using gene ontology enrichment analysis (Supplementary Fig. 1c). We found that Mfn2 interacted with the expected mitochondrial protein partners but, we also identified non-mitochondrial protein partners. The non-mitochondrial proteins interacting with Mfn2 included cadherin binding partners and cell–cell adhesion partners such as KRT18, ENO1, or MACF1 (Supplementary Fig. 1c). In order to understand the functional significance of this intriguing finding, we focused on studying the role of non-mitochondrial Mfn2.

Next, we investigated whether Mfn2 co-localizes with cadherin proteins in ECs using three-dimensional structured illumination microscopy (3D-SIM) which provides high spatial resolution to resolve individual protein complexes[18,19]. 3D-SIM demonstrated the presence of non-mitochondrial Mfn2 complexes, especially at the plasma membrane where Mfn2 co-localized with the endothelial adherens junction (AJ) protein VE-cadherin (Fig. 1a±c and Supplementary Fig. 1d, e). As expected, the majority of Mfn2 co-localized with the mitochondrial membrane protein Tom20 (R2 in Fig. 1a, b) as previously reported[20]. However, we found that Mfn2 additionally co-localized with the endothelial AJ protein VE-cadherin but not with Tom20, which indicated that non-mitochondrial Mfn2 was present at the AJs (R1 in Fig. 1a, b). Manders' overlap coefficient indicates that 54% of Mfn2 (M1 = 0.54 ± 0.045) co-localizes with VE-cadherin and 33% of VE-cadherin (M2 = 0.33 ± 0.055) co-localizes with Mfn2 at AJs area (Fig. 1c). Confocal microscopy experiments involving the expression of GFP-Mfn2 independently confirmed the presence of non-mitochondrial Mfn2 (not co-localizing with Tom20) (R1 in Fig. 1d, e), indicative of an unrecognized non-mitochondrial role consistent with the proteomic analysis (Supplementary Fig. 1c). In contrast, its homolog Mfn1 did not co-localize with VE-cadherin at the plasma membrane (R1 in Fig. 1f, g, R1 in Supplementary Fig. 1h, i) and instead always co-localized with the Tom20 (R2 in Fig. 1f, g, R1 in Supplementary Fig. 1h, g) as evidenced by the low Manders' overlap coefficient (M1 = 0.0019 ± 0.0246 for Mfn1, M2 = 0.0074 ± 0.0031 for VE-cadherin) (Fig. 1h) and low Spearman coefficient (R = 0.195) (Supplementary Fig. 1j) between Mfn1 and VE-cadherin at AJs.

To understand the specific roles of Mfn2 in AJ protein complexes, we next investigated potential binding partners of Mfn2 within AJs in HLMVECs. Total Mfn2 proteins were immunoprecipitated and followed by immunoblotting with antibodies for AJ complex proteins VE-cadherin or β-catenin. Mfn2 significantly interacted with VE-cadherin and β-catenin in resting ECs (Fig. 1i, j). To verify the specificity of the Mfn2 interactions at AJs, Mfn2 was depleted in HLMVECs with a doxycycline inducible lentiviral Mfn2 shRNA. Mfn2 depletion (Mfn2-KD) decreased the interaction of Mfn2 with VE-cadherin and β-catenin (Supplementary Fig. 1k). Importantly, these biochemical approaches independently confirmed that Mfn2 interacts with the endothelial AJ proteins VE-cadherin and

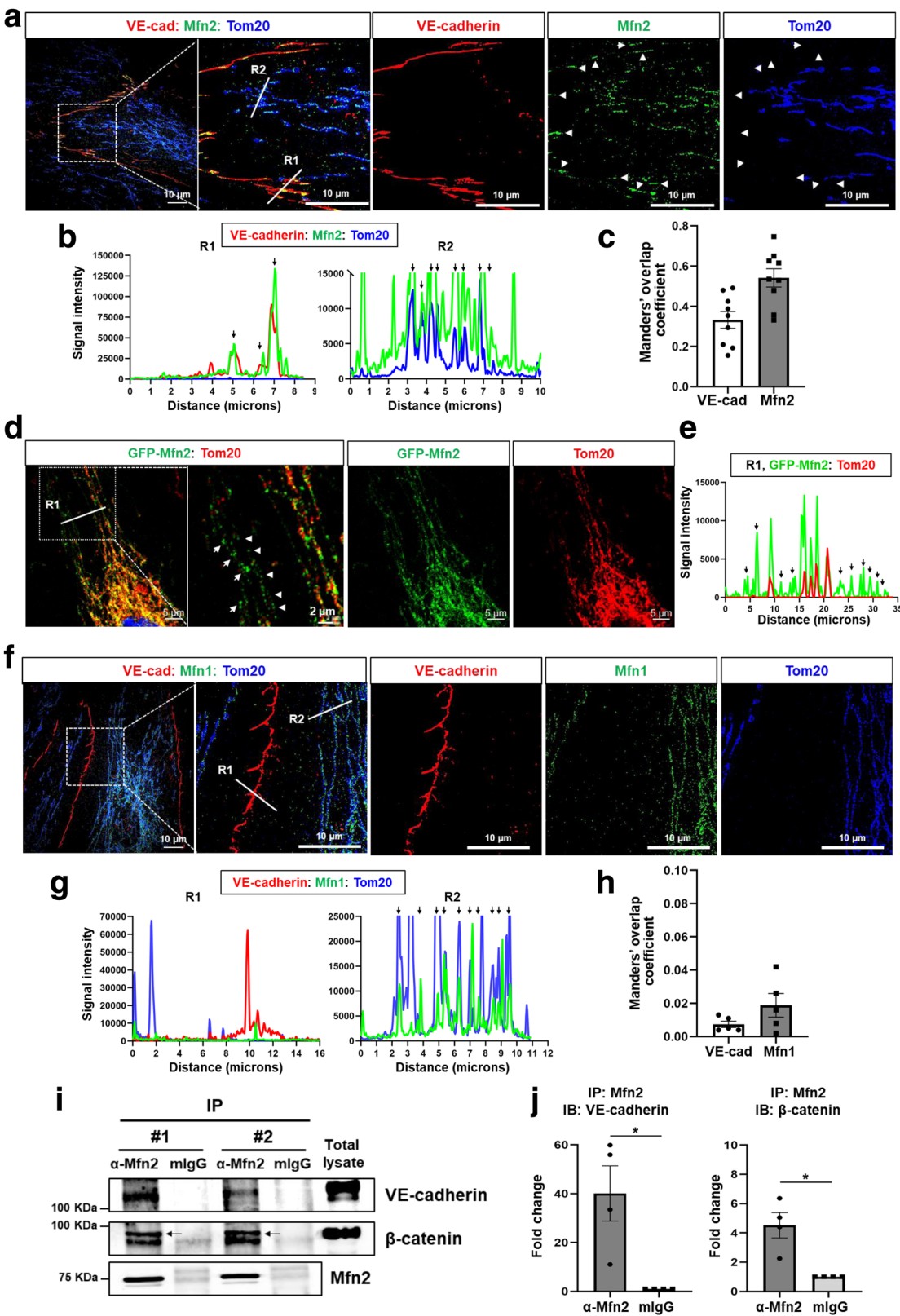

β-catenin at the plasma membrane, consistent with the 3D-SIM findings (Fig. 1a).

**Mfn2 localized at the plasma membrane promotes endothelial barrier integrity during endothelial homeostasis.** To determine whether the presence of Mfn2 at AJs affects AJ function as

manifested by endothelial barrier integrity, we next investigated endothelial barrier function in control and Mfn2-depleted HLMVECs. We confirmed that our Mfn2 depletion strategy was specific to Mfn2 and did not affect Mfn1 levels (Supplementary Fig. 2a, b). Mfn2 depletion disrupted adherens junctions (AJs) as visualized by VE-cadherin and β-catenin immunostaining (Fig. 2a), whereas Mfn1 depletion did not affect the endothelial

**Fig. 1 Non-canonical localization of Mfn2 at the plasma membrane in endothelial cells.** Confluent HLMVECs were fixed with 4% PFA for 10 min and permeabilized with 0.25% Triton X-100 for 10 min. **a** Mfn2 cellular localization was examined by Mfn2 immunostaining with co-staining of the mitochondrial membrane protein Tom20 and the AJ protein VE-cadherin. The images were taken using three-dimensional illumination microscopy (SIM, DeltaVision OMX, Olympus 60X/1.42 NA Plan Apo objective). Images represent maximum intensity projection of 15–17 Z-planes (125 nm step size) acquired in full frame (1024 × 1024 pixel) structured illumination mode. Mfn2, Tom20, and VE-cadherin were pseudocolored with green, blue, and red, respectively. The white arrow heads represent non-mitochondrial Mfn2 at the AJs along the plasma membrane. **b** Co-localization of VE-cadherin, Mfn2, and Tom20 was analyzed by comparing the fluorescence intensity for each protein. The white lines, R1 and R2 of (**a**), indicate the area where the distance between Mfn2 and VE-cadherin, or Mfn2 and Tom20 was analyzed using ImageJ, respectively. The black arrow heads represent their co-localized regions. **c** The co-localization of Mfn2 and VE-cadherin was statistically estimated by Manders' overlap coefficient analysis. The images are representative of $n = 9$ biologically independent samples. Data are mean values ± SEM. $p = 0.0048$ by paired, two-tailed t-test. **d** HLMVECs expressing GFP-Mfn2 were used to determine localization of Mfn2 and Tom2 by immunostaining with a specific antibody for Tom20 using confocal microscopy (Zeiss LSM880, Plan 1.46NA, ×63 magnification). Green color indicates GFP-Mfn2 and red color indicates Tom20. The white arrow heads represent non-mitochondrial Mfn2 in the cytosol. **e** Co-localization analysis of GFP-Mfn2 and Tom20 at R1 of (**d**) using ImageJ. The black arrows represent non-mitochondrial Mfn2. The images are representative of at least 3 independent experiments. **f** Mfn1 cellular localization was examined by Mfn1 immunostaining with co-staining of Tom20 and VE-cadherin using 3-D SIM. Images represent maximum intensity projection of 15–17 Z-planes (125 nm step size) acquired in full frame (1024 ×1024 pixel) structured illumination mode. Mfn1, Tom20, and VE-cadherin were pseudocolored with green, blue, and red, respectively. **g** Co-localization analysis of VE-cadherin and Mfn1 at R1 and R2 of (**f**) using ImageJ. R1 and R2 was used for analyzing co-localization of Mfn1 and VE-cadherin or of Mfn1 and Tom20, respectively, and black arrows represent their co-localized regions. **h** The co-localization of Mfn1 and VE-cadherin was statistically estimated by Manders' overlap coefficient analysis. The images are representative of $n = 5$ biologically independent samples. Non-significance (ns), $p = 0.1577$ by unpaired, two-tailed t-test. **i** Confluent resting HLMVECs were used to determine interaction of Mfn2 and AJ proteins, VE-cadherin and β-catenin. Mfn2 were immunoprecipitated with a specific antibody followed by Western blotting with VE-cadherin or β-catenin antibodies. #1 and 2 indicate two independent experiments. Uncropped blots can be found in the Source Data file. **j** Quantification of band intensities in (**l**) using Image J. Data are mean values ± SEM for $n = 4$ independent experiments. *$p = 0.0407$ for VE-cadherin, *$p = 0.0261$ for β-catenin by paired, two-tailed t-test.

barrier (Supplementary Fig. 2c), indicating that the observed barrier stabilizing effect was specific for Mfn2. The area and fluorescence intensity of VE-cadherin or β-catenin at AJs were quantified with ImageJ (Fig. 2b and Supplementary Fig. 2d). Next, we used a transendothelial resistance (TER) assay and a FITC-conjugated albumin permeability assay to assess endothelial barrier integrity. Mfn2 depletion significantly decreased transendothelial resistance (TER) (Fig. 2c and Supplementary Fig. 2e). Similarly, the transendothelial permeability for the FITC-conjugated albumin tracer was increased in Mfn2-depleted HLMVEC monolayers (Fig. 2d). Moreover, we examined whether the loss of barrier integrity could be rescued by GFP-Mfn2 overexpression. We designed an Mfn2 shRNA to target 3′-UTR regions of *Mfn2* gene instead of the coding sequences (CDS), thus making GFP-Mfn2 impervious to Mfn2 shRNA. We observed that the increased permeability in Mfn2-KD cells was significantly rescued by overexpressing GFP-Mfn2 as assessed by FITC-conjugated albumin permeability (Fig. 2d). We also assessed the barrier by confocal microscopy and found that the disrupted barrier in Mfn2-depleted ECs was restored after GFP-Mfn2 overexpression (Fig. 2e, f).

It is known that filamentous actin (F-actin) plays a critical role in stabilizing cell–cell contacts at AJs[21]. Our proteomic analysis had shown that Mfn2 interacts with an actin binding protein, MACF1 (Microtubule Actin Crosslinking Factor 1) (Supplementary Fig. 1c). Thus, we examined whether the effects of Mfn2 on the endothelial AJ stability may in part relate to its interaction with F-actin. Control endothelial cells showed a regular F-actin structure, whereas Mfn2-KD ECs clearly demonstrated decreased sites of cell–cell contact and a disassembly of F-actin contact points using LifeAct-EGFP live cell imaging of actin filaments (Supplementary Fig. 2f). Furthermore, we found that F-actin co-localization with VE-cadherin was dependent on Mfn2 (Supplementary Fig. 2g, h). These results suggest that Mfn2 may help anchor F-actin structures at AJs and thus promote endothelial barrier integrity.

In addition, we investigated the possibilities that our observed effects of Mfn2 depletion on the loss of endothelial barrier integrity may reflect a form of generalized cellular stress that would increase cell death, oxidative stress, or reduce cell proliferation. We found that Mfn2 depletion in homeostatic ECs did not decrease cell proliferation or induce apoptosis (Supplementary Fig. 3a–f). Furthermore, Mfn2 depletion did not increase mitochondrial ROS production (Supplementary Fig. 3g, h) and also had no significant effect on mitophagy (Supplementary Fig. 3i, j), ER stress (Supplementary Fig. 3k), and total AJs protein levels in homeostatic ECs (Supplementary Fig. 2l). Next, we examined whether changes in the mitochondrial morphology could affect EC barrier integrity, as Mfn2 is a key regulator of mitochondrial fusion and therefore any effects on EC barrier integrity may be mediated by changing mitochondrial dynamics. We used siRNA to specifically deplete the mitochondrial fission mediator Drp1 or the mitochondrial fusion mediators Opa1, Mfn1, and Mfn2 (Supplementary Fig. 3m, n). Even though the siRNA depletions were sufficient to modify mitochondrial network structure, as seen in the increase of mitochondrial fragmentation following Opa1 depletion or the increase in mitochondrial elongation with Drp1 depletion, EC barrier integrity was only significantly decreased with Mfn2 depletion suggesting that the observed Mfn2 effects on EC barrier integrity were not primarily related to the Mfn2 role in mitochondrial fusion. Taken together, these data suggest that Mfn2 binds to AJ complexes at the plasma membrane where it specifically stabilizes the endothelial barrier and regulates F-actin filament structures.

**Endothelial-specific deletion of Mfn2 increases inflammatory injury in vivo.** We next studied whether Mfn2 regulates the endothelial barrier integrity in vivo and used a tamoxifen-inducible EC-specific conditional Mfn2 knockout (Mfn2[EC−/−]) mice (Supplementary Fig. 4a, b). First, we investigated mRNA or protein expression levels of Mfn2 in whole lungs from Mfn2[fl/fl] (littermate controls) and Mfn2[EC−/−] mice. The mRNA levels of Mfn2 showed about 50% knockdown efficiency in whole lungs from Mfn2[EC−/−] mice (Supplementary Fig. 4b). To verify specific deletion of Mfn2 in ECs, we examined protein levels of Mfn2 in ECs isolated from the lungs of Mfn2[fl/fl] and Mfn2[EC−/−] mice. ECs from Mfn2[EC−/−] mice demonstrated greater than 80% reduction of Mfn2 levels compared to ECs from Mfn2[fl/fl] (Fig. 3a, b). Next, we examined the functional role of endothelial Mfn2 in the regulation of in vivo vascular permeability. Mfn2[EC−/−] and

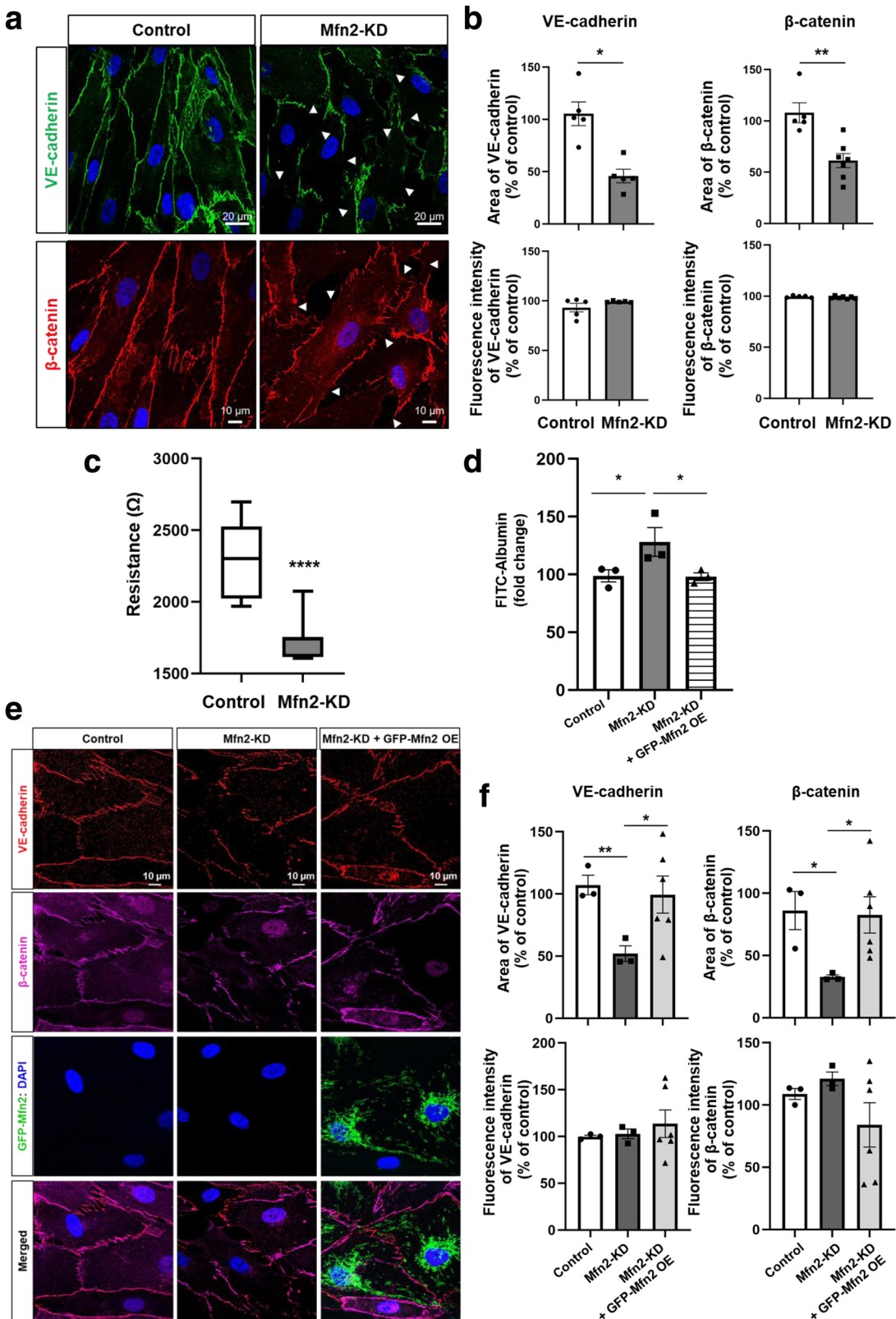

Mfn2$^{fl/fl}$ mice were injected intravenously with the Evans blue-albumin dye to assess lung vascular permeability[22]. The lungs from Mfn2$^{EC-/-}$ mice demonstrated significantly higher leakiness than lungs from Mfn2$^{fl/fl}$ control mice (Fig. 3c, d). We also examined the expression levels of pro-inflammatory genes in the lungs of Mfn2$^{fl/fl}$ or Mfn2$^{EC-/-}$ mice. As shown in Fig. 3e, the expression of *IL-β*, *IL-6*, *TNFα,* or *IFNγ* genes was significantly

increased in lungs of Mfn2$^{EC-/-}$ mice than in those of Mfn2$^{fl/fl}$ control mice (Fig. 3e).

Furthermore, the lungs from Mfn2$^{EC-/-}$ mice demonstrated significant accumulation of immune cells in the proximity of venous blood vessels but not arterial blood vessels when compared to the lungs of control Mfn2$^{fl/fl}$ mice (Fig. 3f). Finally, to determine the phenotype of infiltrating immune cells in the

**Fig. 2 Mfn2 at the plasma membrane promotes endothelial barrier integrity.** HLMVECs were infected with doxycycline inducible lenti-Mfn2 shRNA virus for 48 h and treated with doxycycline for 72 h to deplete endogenous Mfn2 (Mfn2-KD). DMSO was used as a vehicle for control ECs. **a** Barrier integrity of control and Mfn2-KD ECs was examined by VE-cadherin and β-catenin immunostaining using confocal microscopy (Zeiss LSM880, Plan 1.46NA, ×63 magnification). **b** The levels of VE-cadherin and β-catenin at the cell surface in (**a**) were determined by analyzing area (upper panel) or fluorescence intensity (lower panel) using ImageJ and represented as % of control. Data are mean values ± SEM for $n = 5$ biologically independent samples. *$p = 0.0184$, **$p = 0.0024$ vs control by paired or unpaired, two-tailed t-test. **c** EC barrier resistance (Ω) of control and Mfn2-KD ECs was measured with TER. The data are presented without normalizing to show baseline difference. Data are presented with box and whiskers plot and whiskers are Min to Max. $n = 5$ biologically independent experiments (TER was measured every 8 s for 40 min). ****$p = 0.0001$ vs control by paired, two-tailed t-test. **d** Control and Mfn2-KD ECs were grown on transwell plates (0.4 μm pore) and treated with a FITC-albumin tracer (0.5 mg/mL) for 4 h. For rescue experiments, Mfn2-KD ECs were overexpressed with GFP-Mfn2 and the cells (Mfn2-KD + GFP-Mfn2 OE) were examined by permeability assay. The concentrations of FITC-albumin tracer were determined in the lower chamber media at excitation 488 nm and emission 520 nm. The data are presented by normalizing to the baseline. Data are mean values ± SEM for three independent experiments. *$p = 0.0298$ for control vs Mfn2-KD, *$p = 0.0399$ for Mfn2-KD vs Mfn2-KD + GFP-Mfn2 OE by unpaired, one-tailed t-test. **e** Barrier integrity of confluence control, Mfn2-KD ECs, and Mfn2-KD + GFP-Mfn2 OE ECs was examined by VE-cadherin and β-catenin immunostaining using confocal microscopy (Zeiss LSM7700, Plan 1.46NA, ×63 magnification). **f** The levels of VE-cadherin and β-catenin at the cell surface in (**e**) were determined by analyzing area (upper panel) or fluorescence intensity (lower panel) using ImageJ and represented as % of control. Data are mean values ± SEM for $n = 3–6$ biologically independent samples. **$p = 0.0053$ for area of VE-cadherin in control vs Mfn2-KD, *$p = 0.0211$ for area of VE-cadherin in Mfn2-KD vs Mfn2-KD + GFP-Mfn2 OE, *$p = 0.0256$ for area of β-catenin in control vs Mfn2-KD, *$p = 0.0354$ for area of β-catenin in Mfn2-KD vs Mfn2-KD + GFP-Mfn2 OE by unpaired, two-tailed t-test.

lungs of Mfn2$^{EC-/-}$ mice, we generated a single cell suspension of the whole lung tissues obtained from Mfn2$^{fl/fl}$ and Mfn2$^{EC-/-}$ mice[23], and performed flow cytometry after staining for cell type-specific markers. We found that the lungs of Mfn2$^{EC-/-}$ mice were significantly enriched for myeloid cells such as interstitial macrophages (IMs) and neutrophils (Neu), as well as lymphoid B + T cells and natural killer T (NKT) cells when compared to the lungs of Mfn2$^{fl/fl}$ mice (Fig. 3g, h). However, B cells and T cells were decreased in the lungs from Mfn2$^{EC-/-}$ mice, suggesting an activation of the innate immune response in resting lung endothelial cells upon endothelial Mfn2 deletion (Supplementary Fig. 4c, d). These data indicate that endothelial Mfn2 is a critical regulator of the endothelial barrier integrity during homeostasis and that its absence induces spontaneous vascular leakiness as well as inflammation.

**Mfn2 interaction with β-catenin is enhanced following their disassociation from adherens junctions.** After establishing a key role of Mfn2 in stabilizing the endothelial barrier during homeostatic conditions, we next examined the role of Mfn2 during inflammation because inflammatory stimulation promotes AJ disassembly[24]. HLMVECs were stimulated with TNFα and the AJ complex proteins VE-cadherin or β-catenin were immuno-precipitated, followed by immunoblotting. We found that inflammatory stimulation increased the interaction between Mfn2 and β-catenin, but not between Mfn2 and VE-cadherin (Fig. 4a, b and Supplementary Fig. 4e, f). We confirmed these results by using a GFP-tagged Mfn2 construct that was expressed in ECs and found a six-fold increase in the interaction between Mfn2 and β-catenin following inflammatory activation with TNFα (Fig. 4c, d). We also used an in situ proximity ligation assay (PLA), which creates a spatial fluorescent signal within a 30–40 nm maximum distance and can thus establish close proximity interactions between protein partners[25,26]. The known interaction between VE-cadherin and β-catenin was used as a positive control. The interaction between VE-cadherin and β-catenin (red dots) was present under homeostatic baseline conditions but rapidly decreased following TNFα stimulation (Fig. 4e, f). However, the binding of Mfn2 and β-catenin (green dots) was markedly increased in the cytosol (Fig. 4g, h). Moreover, EC barrier impairment in Mfn2-KD ECs was further increased by TNFα stimulation, and rescued by overexpressing GFP-Mfn2 (Supplementary Fig. 4g). Based on multiple lines of inquiry, we concluded that Mfn2 disassociates from VE-cadherin and there is

a concomitant increase of Mfn2 interaction with β-catenin in the cytosol following inflammatory stimulation.

**TNFα-induced ROS increase the binding of Mfn2 to β-catenin.** We next investigated the mechanism by which inflammatory activation with TNFα could affect the interaction between Mfn2 and β-catenin. ROS are key mediators of inflammatory signaling by increasing reversible oxidative modifications[8]. As TNFα has been shown to increase ROS production[7], we posited that ROS-induced cysteine modifications may post-translationally regulate interactions of Mfn2 with partner proteins. ROS production increased within 15 min of inflammatory stimulation with TNFα (Fig. 5a). Exogenous ROS ($H_2O_2$) increased the interaction of Mfn2 and β-catenin (Fig. 5b), consistent with the notion that Mfn2 modified by inflammation-induced ROS promotes the interaction of Mfn2 with cytosolic β-catenin when both are disassociated from AJ complexes, but that Mfn2 no longer interacts with VE-cadherin. Moreover, the complex of Mfn2 and β-catenin exhibited band shifts that were dependent on Mfn2 in a non-reducing SDS-PAGE gel (Supplementary Fig. 5a) and decreased in Mfn2-KD ECs in reducing SDS-PAGE gels (Supplementary Fig. 5b). These data suggest that their interaction may be mediated via disulfide bond formation that was dependent on the presence of ROS. Then, we performed a rescue experiment using the ROS scavenger N-Acetyl-L-cysteine (NAC)[27] which reduces disulfide bond formation[28]. As shown in Fig. 5c, d, the increased interaction of Mfn2 and β-catenin by $H_2O_2$ was reversed by NAC treatment. These data suggest that during inflammation, ROS mediate the interaction of Mfn2 and β-catenin. Moreover, we investigated whether the complex formation of Mfn2 and β-catenin was associated with cysteine sulfenylation, a key initial step for cysteine oxidation. To examine whether TNFα induced-ROS modulate cysteine sulfenylation of Mfn2 and β-catenin, TNFα stimulated-ECs were lysed with lysis buffer containing DCP-Bio1, a cell permeable biotin-labeled Cys-OH trapping probe (DCP-Bio1)[29] to capture all sulfenylated-proteins. The captured proteins were used to determine sulfenylation of Mfn2 or β-catenin by Western blotting with their specific antibodies. Importantly, sulfenylation of both Mfn2 and β-catenin was significantly increased 1 h after TNFα stimulation and then subsequently decreased in a time-dependent manner (Fig. 5e, f). We then addressed the role of sulfenylation in the binding of Mfn2 and β-catenin. Control and Mfn2-depleted ECs were stimulated with 500 μM $H_2O_2$ for 30 min and subjected to a DCP-Bio1 assay. $H_2O_2$ treatment of control ECs induced cysteine sulfenylation of

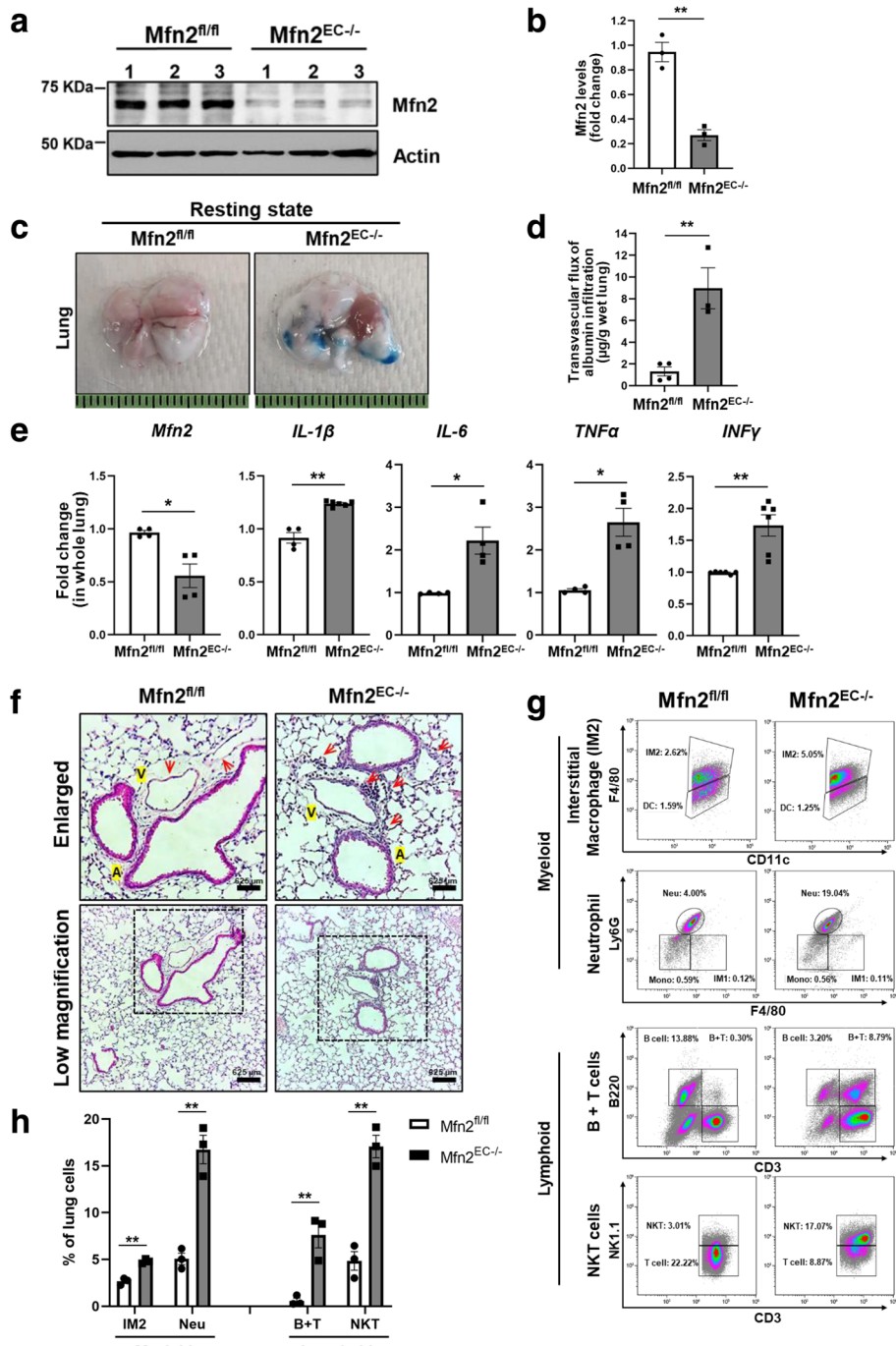

**Fig. 3 Increase in lung vascular permeability following endothelial-specific genetic deletion of Mfn2. a, b** ECs were isolated from the lungs of Mfn2$^{fl/fl}$ or Mfn2$^{EC-/-}$ mice using CD31 antibody. Mfn2 knockdown efficiency was determined by Western blotting (**a**) and protein levels for Mfn2 were quantified using ImageJ (**b**). Uncropped blots can be found in the Source Data file. Data are mean values ± SEM for n = 3 mice. **p = 0.0017 by unpaired, two-tailed t-test. **c** Mfn2$^{fl/fl}$ and Mfn2$^{EC-/-}$ mice in resting condition were i.v. injected with Evans blue albumin (40 mg/mL). After 45 min, the mice were perfused with PBS and lung permeability was evaluated by measuring Evans blue albumin contents in lung. **d** Evans blue albumin contents in lungs was measured at OD620 and the transvascular flux of albumin infiltration is presented (in µg) after normalizing by the wet lung weight (in g). Data are mean values ± SEM for at least n = 3–4 mice. **p = 0.0056 by unpaired, two-tailed t-test. **e** Lungs from Mfn2$^{fl/fl}$ and Mfn2$^{EC-/-}$ mice in homeostatic conditions were used to evaluate the mRNA levels of pro-inflammatory genes such as IL-1β (n = 4-6), IL-6 (n = 4), TNFα (n = 4), and IFNγ (n = 6) by qRT-PCR. Data are mean values ± SEM for at least n = 4 mice. *p = 0.0208 for Mfn2, **p = 0.035 for IL-1β, *p = 0.0314 for IL-6, *p = 0.0127 for TNFα, **p = 0.0075 for IFNγ by paired, two-tailed t-test. **f** Lungs from Mfn2$^{fl/fl}$ (n = 3) and Mfn2$^{EC-/-}$ (n = 3) mice in homeostatic conditions were used to evaluate infiltration of inflammatory cells. The sections from paraffin fixed lungs were represented by H&E staining. Biological replicates (n = 3 mice) from each group showed similar results. **A**: artery, **V**: vein. Red arrow heads represent infiltrated inflammatory cells. **g–h** Lungs from Mfn2$^{fl/fl}$ and Mfn2$^{EC-/-}$ mice in homeostatic conditions were homogenized and inflammatory cells were stained with indicated specific antibodies by following FACS analysis (**g**). **h** The inflammatory cells were represented by percent (%) of whole lung cells in (**g**). Data are mean values ± SEM for n = 3 mice. **IM2**: Interstitial macrophages 2, **Neu**: neutrophil, **B**: B cells, **T**: T cells, **NKT**: Natural killer T cell. **p = 0.0011 for IM2, **p = 0.0018 for Neu, **p = 0.0073 for B + T, **p = 0.0014 for NKT by unpaired, two-tailed t-test. The sequential gating strategies were presented at Supplementary Fig. 4d. All Mfn2$^{fl/fl}$ and Mfn2$^{EC-/-}$ mice received tamoxifen and were rested for one month before experiments.

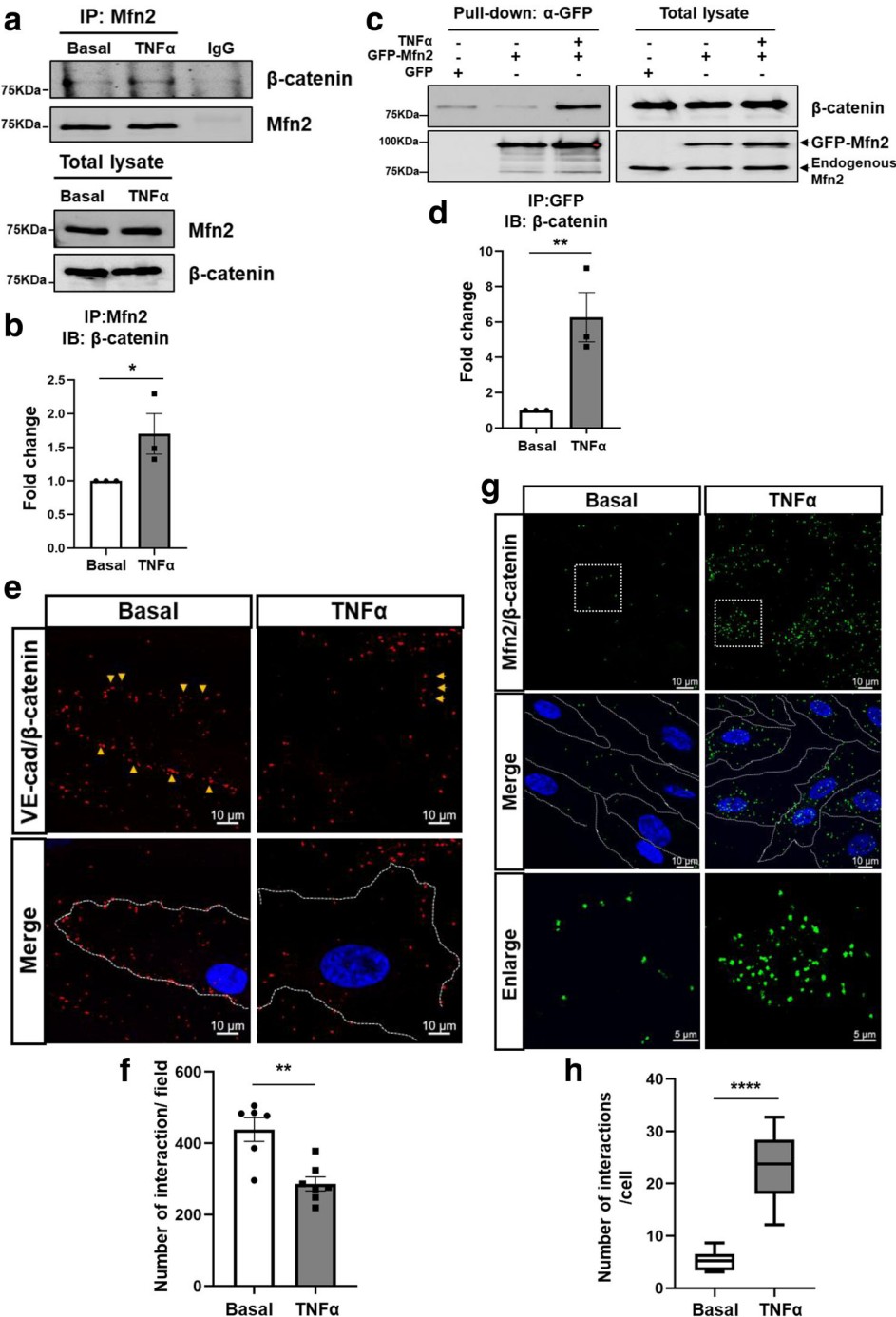

**Fig. 4 Mfn2 binding to the AJ complex protein β-catenin is increased following inflammatory stimulation.** Confluent HLMVECs were treated with TNFα (10 ng/mL) for 6 h. **a** Mfn2 was immunoprecipitated with Mfn2 specific antibody followed by Western blotting for β-catenin or Mfn2 antibodies. Uncropped blots can be found in the Source Data file. **b** The band intensities for Mfn2 and β-catenin interactions in (**a**) were quantified with ImageJ. Data are mean values ± SEM for three independent experiments. *$p = 0.0403$ by unpaired, one-tailed t-test. **c** HLMVECs expressing GFP or GFP-Mfn2 were stimulated with TNFα (10 ng/mL) for 6 h. The GFP-Mfn2 was immunoprecipitated with GFP-trap magnetic agarose followed by Western blotting for β-catenin and Mfn2 antibodies. Uncropped blots can be found in the Source Data file. **d** The band intensities of the Western blot were quantified with ImageJ. Data are mean values ± SEM for three independent experiments. *$p = 0.0097$ by unpaired, one-tailed t-test. **e–h** Proximity ligation assay (PLA). **e** The interaction of endogenous VE-cadherin and β-catenin in HLMVECs was examined using mouse VE-cadherin antibody and rabbit β-catenin antibody for positive control. The red dots show an interaction between VE-cadherin and β-catenin. **f** Quantification of the number of interactions (red dots) per field in (**e**). Data are mean values ± SEM from $n = 6$ for basal and $n = 7$ for TNFα treatment in independent biological replicates samples. **$p = 0.0018$ by paired, two-tailed t-test. **g** The interaction of endogenous Mfn2 and β-catenin in HLMVECs was examined with mouse Mfn2 antibody and rabbit β-catenin antibody. The green dots show interaction between Mfn2 and β-catenin, and DAPI (blue) indicates nuclei. **h** Quantification of the number of interactions (green dots) per cell in (**g**). Data are presented with box and whiskers plot and whiskers are Min to Max. $n = 8$ for basal and $n = 14$ for TNFα. ****$p < 0.0001$ by paired, two-tailed t-test. The images in (**e**) and (**g**) are representative from at least six independent biological replicates.

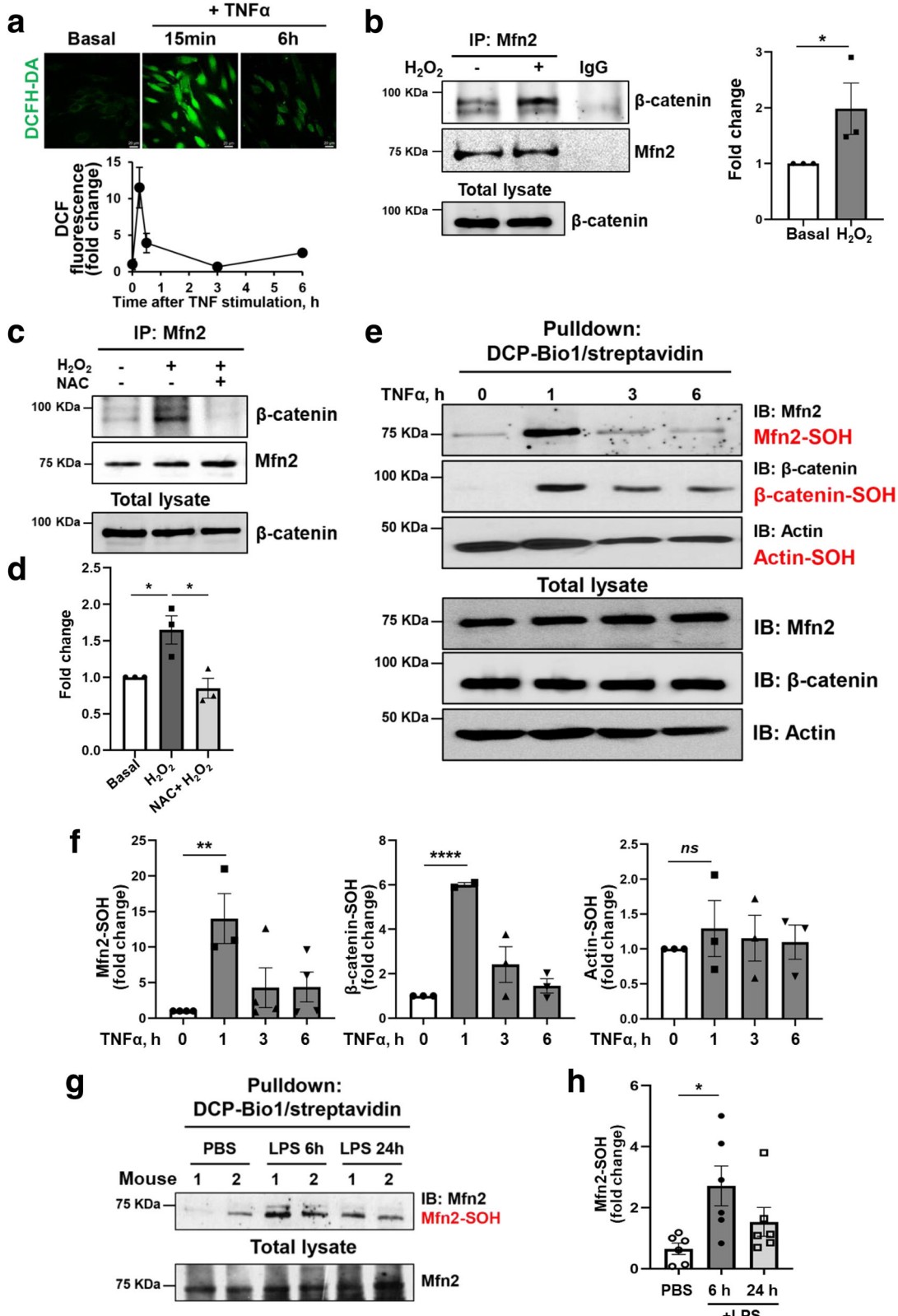

Mfn2 and β-catenin, whereas Mfn2 depletion of ECs resulted in the inhibition of β-catenin sulfenylation (Supplementary Fig. 5c).

To investigate whether sulfenylation of Mfn2 also occurs during inflammation in vivo, we used the experimental model of endotoxemia which induces profound inflammatory injury. C57/BL6 mice were intraperitoneally injected with the bacterial endotoxin lipopolysaccharide (8 mg/kg sublethal LPS) or PBS

(control) and we monitored Mfn2 sulfenylation at 6 h and 24 h in whole lungs. LPS induced inflammation significantly increased Mfn2 sulfenylation at 6 h (Fig. 5g, h), indicating that Mfn2 sulfenylation is a post-translational modification that also occurs in vivo. However, we found that Mfn2 depletion or TNFα stimulation did not induce phosphorylation at serine 33 and 37 residues of β-catenin which are known to promote β-catenin

**Fig. 5 TNFα induced ROS production mediates the interaction of Mfn2 and β-catenin. a** HLMVECs were stimulated with TNFα (10 ng/mL) for indicated durations (0, 15 min, 3 h, and 6 h). TNFα-induced total ROS production was measured using a DCFH-DA probe which detects intracellular $H_2O_2$. The ROS images in *upper panel* were taken with confocal microscope (Zeiss LSM880, Plan 1.45NA, ×63 magnification). The relative fluorescence of DCF was quantified with ImageJ (*lower panel*) and presented fold change to basal. Data are mean values ± SD from $n = 4$. **b** *Left panel*, HLMVECs were stimulated with $H_2O_2$ (500 μM) for 30 min and immunoprecipitated with Mfn2 antibody followed by Western blotting with β-catenin antibody under reducing conditions (with β-mercaptoethanol, ME). Mouse normal IgG was used as a negative control for immunoprecipitation. Uncropped blots are provided in the Source Data file. *Right panel*, the quantification of band intensities for Mfn2 and β-catenin interactions with ImageJ. Data are mean values ± SEM for three independent experiments. *$p = 0.050$ by unpaired, one-tailed t-test. **c** HLMVECs were pretreated with 20 mM NAC for 30 min. The cells were stimulated with $H_2O_2$ (500 μM) for 30 min and immunoprecipitated with Mfn2 antibody followed by Western blotting with β-catenin antibody. Uncropped blots are in the Source Data file. **d** Quantification of band intensity for Mfn2 and β-catenin interactions in (**c**) with ImageJ. Data are mean values ± SEM for three independent experiments. *$p = 0.0287$ for Basal vs $H_2O_2$, *$p = 0.0281$ for $H_2O_2$ vs $H_2O_2$ + NAC by unpaired, two-tailed t-test. **e** Confluent HLMVECs were stimulated with TNFα (10 ng/mL) for the indicated time course (0, 1, 3, or 6 h) and lysed with lysis buffer containing DCP-Bio1. The sulfenylated (DCP-Bio1 conjugated) proteins were pull downed with streptavidin and sulfenylation of Mfn2 or β-catenin was determined by Western blotting with their specific antibodies. Actin was used as a negative control for TNFα induced sulfenylation. Uncropped blots are provided in the Source Data file. **f** Quantification of sulfenylation levels of Mfn2, β-catenin, and actin in (**e**) with ImageJ. Data are mean values ± SEM for 3–4 independent experiments. **$p = 0.0068$ for Mfn2-SOH in TNFα 0 h vs 1 h, ****$p < 0.0001$ for β-catenin-SOH in TNFα 0 h vs 1 h, *ns*: not significant ($p = 0.5061$) for Actin-SOH in TNFα 0 h vs 1 h by unpaired, two-tailed t-test. **g, h** C57/BL6 control mice were administrated with PBS or sublethal LPS (8 mg/kg i.p.) and sacrificed after 6 h or 24 h. The whole lungs after PBS perfusion were used for DCP-Bio1 assay and followed by streptavidin pulldown and Western blotting (**g**). Uncropped blots are provided in the Source Data file. **h** Quantification of sulfenylated Mfn2 in the lungs shown in (**g**) with ImageJ. The levels of Mfn2-SOH were normalized by the total Mfn2 loading amount and further normalized with controls (PBS), and are presented as fold change. Data are mean values ± SEM for $n = 6$ mice for each condition. *$p = 0.0126$ for PBS vs LPS 6 h, $p = 0.1142$ for PBS vs LPS 24 h by unpaired, two-tailed t-test.

degradation (Supplementary Fig. 5d, e). TNFα (10 ng/mL) stimulation had no effect on total protein levels of AJs complex until 24 h (Supplementary Fig. 5f). Taken together, these data suggest that inflammation-induced Mfn2 sulfenylation increases its interaction with sulfenylated-β-catenin.

**Transcriptional activity of β-catenin is negatively regulated by Mfn2 during inflammation**. To address the potential functional roles of the Mfn2-β-catenin interaction during inflammation, we investigated whether the presence of Mfn2 affects the expression of key pro-inflammatory genes such as Intercellular cell adhesion molecule-1 (*ICAM-1*), Interleukin-6 (*IL-6*), and −18 (*IL-18*). Control and Mfn2-KD ECs were stimulated with TNFα and mRNA expression levels of pro-inflammatory genes were evaluated by quantitative real-time PCR (qRT-PCR). Mfn2 depletion in ECs significantly increased TNFα-induced mRNA levels of *ICAM-1*, *IL-6*, and *IL-18*, indicating that Mfn2 suppresses the TNFα induced pro-inflammatory response in ECs (Fig. 6a, b).

Upon disassociating from AJs complexes at plasma membrane, β-catenin translocates to the nucleus where it acts as a key co-factor for the transcription factor, T-cell factor (TCF) and thereby mediates Wnt signaling[30]. Since we had found that Mfn2 binds β-catenin after inflammation-induced disassociation from AJs, we next investigated whether inflammation affects β-catenin transcriptional activity using a β-catenin luciferase assay. TNFα stimulation significantly increased β-catenin transcriptional activity in a time-dependent manner (Supplementary Fig. 6a). Interestingly, β-catenin transcriptional activity was significantly increased by Mfn2 depletion and further enhanced by TNFα stimulation in ECs (Fig. 6c), thus suggesting that Mfn2 acted as a suppressor of β-catenin-mediated transcriptional activation during inflammation. The exaggerated β-catenin transcriptional activity was reset by restoring Mfn2 expression (Fig. 6d).

We then examined whether β-catenin directly regulates the expression of pro-inflammatory genes. HLMVECs were transfected with siRNA for β-catenin (Supplementary Fig. 6b) and mRNA levels of pro-inflammatory genes were examined with or without TNFα stimulation. The TNFα-induced expression of *ICAM-1*, *IL-β*, or *IL-18* genes was significantly inhibited in β-catenin knockdown ECs (Fig. 6e). Taken together, these data indicate that Mfn2 functions as a suppressor for β-catenin-mediated transcriptional activation during inflammation.

**TNFα triggers nuclear accumulation of Mfn2**. Next, we investigated whether Mfn2 accumulates in the nucleus during inflammation. The precise subcellular localization of Mfn2 was determined by a subcellular fractionation assay as well as by confocal microscopy at baseline and following TNFα stimulation. We first confirmed nuclear translocation of the pro-inflammatory transcription factor NF-kB after TNFα stimulation[29,31] to validate the degree of inflammatory activation induced by TNFα in ECs (Supplementary Fig. 6d). Mfn2 was mainly localized in the cytosolic/mitochondrial fraction but a portion of Mfn2 clearly accumulated in the nuclei at 6 h after TNFα stimulation (Fig. 7a, b), whereas Mfn1 showed no such accumulation (Supplementary Fig. 6d). It came as a surprise because Mfn2 does not have a nuclear localization sequence (NLS) even though the presence of mitochondrial proteins in the nucleus has been recently described[32,33]. To verify nuclear accumulation of Mfn2 during inflammation, we investigated whether exogenously over-expressed GFP-Mfn2 is also found in the nucleus along with endogenous Mfn2 during inflammation using confocal microscopy with three-dimensional Z-stacking. GFP-Mfn2 accumulated in the nucleus along with endogenous Mfn2 after 6 h of TNFα stimulation (Fig. 7c, d). Moreover, it was confirmed by 3-dimensional image analysis using all sections (average 91 sections) and ortho analysis using one section to show co-localization of GFP-Mfn2, Mfn2 (red color), and DAPI (Fig. 7e). We found that β-catenin translocated into the nucleus at 6 h after TNFα stimulation, mirroring the nuclear accumulation pattern we had observed for Mfn2 (Fig. 7f, g and Supplementary Fig. 6d, e). The persistence of the Mfn2/β-catenin interaction in the nucleus by TNFα stimulation was further demonstrated by a proximity ligation assay (PLA) (Fig. 7h). We next examined whether Mfn2 affected nuclear accumulation of β-catenin using confocal microscopy. The TNFα-induced nuclear accumulation of β-catenin was not affected by Mfn2 depletion (Supplementary Fig. 6f, g). Taken together, these results indicate that Mfn2 and β-catenin both accumulate in the nucleus and that nuclear Mfn2 may suppress the transcriptional activity of β-catenin.

**Discussion**
Our goal was to identify binding partners of Mfn2 and potential non-mitochondrial roles of Mfn2 in the endothelium. The non-mitochondrial roles for Mfn2 we identified in the

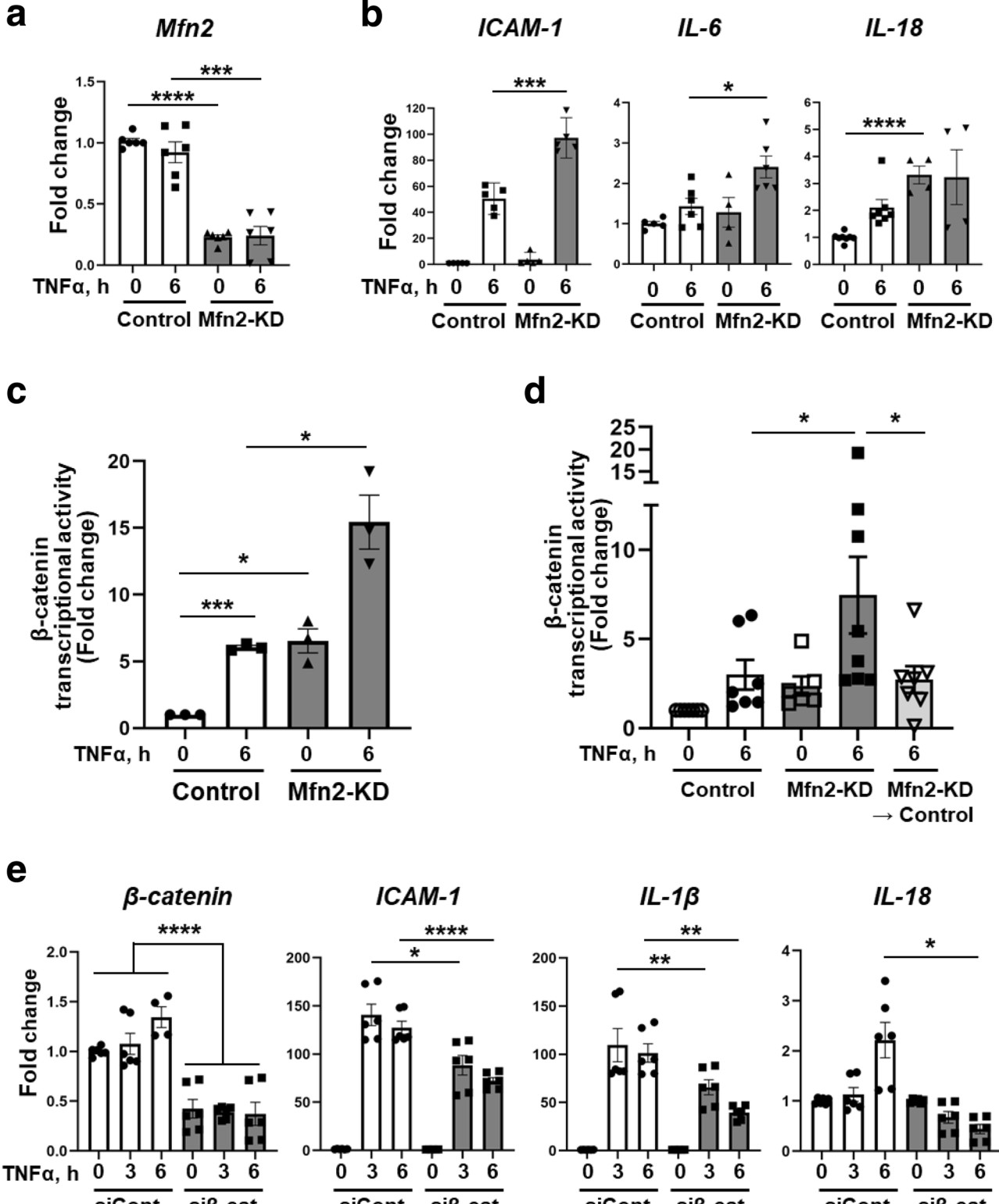

endothelium include: (i) stabilization of endothelial AJ junctions by binding to the AJ proteins complex VE-cadherin and β-catenin under homeostatic conditions, (ii) Sulfenylation of Mfn2 disassociated from the AJ complex during inflammation and accumulation of Mfn2 in the nucleus, and (iii) the binding of Mfn2 to the transcriptional regulator β-catenin and inhibition of β-catenin transcriptional activity during inflammatory activation.

Mfn2 is typically found in the outer mitochondrial membrane and its structural motifs contain a cytosolic N-terminal GTPase domain, a proline-rich region (PR), two coiled-coil heptad-repeat domains (HR1 and HR2), and a transmembrane domain (TM) which allows Mfn2 anchorage in the outer mitochondrial membrane[34]. Mfn1 and Mfn2 are key mediators of mitochondrial fusion but only Mfn2 has a PR domain, thought to be responsible for specific protein–protein interactions[14] suggesting the

**Fig. 6 Mfn2 suppresses β-catenin transcriptional activity which requires TNFα induced pro-inflammatory gene expression. a, b** Control and Mfn2-KD ECs were stimulated with or without TNFα (10 ng/mL) for 6 h and total RNA was extracted. **a** The knockdown efficiency of *Mfn2* mRNA was determined under basal and TNFα stimulation by qRT-PCR with its specific primers. Data are mean values ± SEM for $n = 6$ independent biological replicates. ****$p < 0.0001$ for *Mfn2* in basal control vs Mfn2-KD. **b** The mRNA levels of pro-inflammatory genes, *ICAM-1, IL-6,* and *IL-18* were determined by qRT-PCR with their specific primers. ***$p = 0.0001$ for *ICAM-1* of TNFα 6 h in control vs Mfn2-KD, *$p = 0.016$ for *IL-6* of TNFα 6 h in control vs Mfn2-KD, ****$p < 0.0001$ for *IL-18* in basal control vs Mfn2-KD by unpaired, two-tailed t-test. Data are mean values ± SEM for $n = 4$–6 independent biological replicates. **c** Control and Mfn2-KD ECs were transfected with 1 μg Topflash (β-catenin reporter containing the TCF promoter) and 35 ng of pRL/TK for 48 h. The cells were stimulated with or without TNFα (10 ng/mL) for 6 h. Firefly and renilla-luciferase activity were determined by the dual luciferase reagent assay system and the firefly luciferase activity was normalized by renilla-luciferase activity. Transcriptional activity of β-catenin was presented for fold change by further normalizing with value of control basal (TNFα 0 h). Data are mean values ± SEM for $n = 3$ independent experiments. ***$p = 0.0008$ for control in TNFα 0 h vs 6 h, *$p = 0.0256$ for basal in control vs Mfn2-KD, *$p = 0.0394$ for TNFα 6 h in control vs Mfn2-KD by paired, two-tailed t-test. **d** HLMVECs expressing doxycycline inducible lentiviral Mfn2 shRNA were transfected Topflash and pRL/TK with the same method in (**c**) for 24 h, and treated with or without doxycycline for 72 h. The rescue experimental group (Mfn2-KD → control) was treated with doxycycline for 48 h and further incubated in media without doxycycline for 24 h. The cells were stimulated with or without TNFα (10 ng/mL) for 6 h. Data are mean values ± SEM for $n = 6$–7 biological independent samples. *$p = 0.0447$ for TNFα 6 h in control vs Mfn2-KD, *$p = 0.0356$ for TNFα 6 h in Mfn2-KD vs Mfn2-KD → control by unpaired, one-tailed t-test. **e** HLMVECs were transfected with siRNA of control or β-catenin for 48 h and stimulated with TNFα (10 ng/mL) for 6 h. The mRNA levels of *β-catenin* and pro-inflammatory genes such as *ICAM-1, IL-1β,* and *IL-18* were evaluated by qRT-PCR. Data are mean values ± SEM for $n = 6$ independent biological replicates. ****$p < 0.0001$ for *β-catenin* by one-way ANOVA, *$p = 0.0116$ for *ICAM-1* of TNFα 3 h in siCont vs siβ-cat, ****$p < 0.0001$ for *ICAM-1* of TNFα 6 h in siCont vs siβ-cat, **$p = 0.004$ for *IL-1β* of TNFα 3 h in siCont vs siβ-cat, **$p = 0.0025$ for *IL-1β* of TNFα 3 h in siCont vs siβ-cat, *$p = 0.0419$ for *IL-18* of TNFα 6 h in siCont vs siβ-cat by unpaired, two-tailed t-test.

possibility for additional roles beyond mitochondrial fusion for Mfn2. In line with this, it has been recently reported that the mitochondria–ER–cortex anchor (MECA) interacts directly with mitochondria and the plasma membrane via core protein component, Num1 in budding yeast[35]. Moreover, it has been appreciated that many organelles communicate by using molecular tethers[36] and the function of the mitochondria–plasma membrane contact extends beyond the mitochondrion itself[35,37].

Our proteomic analysis revealed unexpected Mfn2 binding partners located outside of the mitochondria such as cell junction proteins. We took advantage of super-resolution microscopy which resolves individual protein complexes[18,19] to validate the extra-mitochondrial localization of Mfn2. Super-resolution imaging identified the expected mitochondrial localization of the bulk of Mfn2 but also clearly visualized Mfn2 localization at the plasma membrane. Biochemical immunoprecipitation assays identified the plasma membrane binding partners of Mfn2 which included the adherens junction proteins VE-cadherin and β-catenin in homeostatic ECs. Loss of Mfn2 resulted in the disruption of the endothelial barrier, thus indicating that Mfn2 is not only localized at the junctions but also plays a functional role by stabilizing barrier integrity. We also found that Mfn2-depleted ECs showed an impaired F-actin structure which is important for cell–cell interaction. Importantly, during inflammatory stimulation, Mfn2 was disassociated from the junctions and this likely contributes to the disruption of barrier integrity that is typically observed during inflammation[3,38]. It is plausible that Mfn2 may play a role as an anchor between AJs proteins at plasma membrane and F-actin at cytosol to maintain EC barrier integrity in homeostatic ECs. We also did not find any broad effects on cell stress, cell death or cell proliferation following Mfn2 deletion, which suggested that our observations were most likely due to a specific role for Mfn2 in AJs and endothelial barrier integrity that is independent of generalized cellular health functions of Mfn2.

We next investigated the mechanism by which Mfn2 disassociated from the adherens junctions during inflammation. Post-translational modifications are regulatory switches which modify the activity of proteins, and oxidation is one of the most frequently occurring post-translational modification[39]. Especially during inflammation, ROS levels can acutely increase and result in oxidative modification of proteins[6]. Cysteine (Cys) is an amino acid that is susceptible to several types of oxidative post-translational modification including sulfenylation, disulfide

formation, S-glutathionylation, and S-nitrosylation[9,40]. Oxidative modifications of proteins are critical mediators of compartmentalized ROS signaling[41]. Recently, it has been shown that increasing concentrations of xanthine oxidase, a cytosolic source of ROS inhibits Mfn2 activity by inducing disulfide linked oligomerization via oxidation of C-terminal Cys 684, 700 residues in vitro[42,43]. However, it is unknown whether redox-dependent modifications of Mfn2 could affect its non-mitochondrial roles as a stabilizer of AJ protein complexes at the plasma membrane in ECs. We found that TNFα-induced ROS increased Mfn2 sulfenylation in ECs. Our results suggest that the sulfenylation step might be required for the disassociation of Mfn2 from adherens junctions and that Mfn2 sulfenylation may constitute a form of post-translational regulation of Mfn2 activity during inflammation. Future studies could identify specific cysteine residues that serve as sulfenylation targets and whether such sulfenylation would also impact other aspects of Mfn2 function such as its GTPase activity and mitochondrial fusion.

Interestingly, we also found Mfn2 presence in the nucleus during inflammatory activation. This surprising nuclear localization of Mfn2 was independently confirmed by several different approaches including biochemical subcellular fractionation, immunofluorescence with 3D analysis, and proximity ligation assays. We also confirmed TNFα-induced accumulation of Mfn2 using exogenously overexpressed GFP-Mfn2. However, there are important questions regarding the mechanisms of Mfn2 nuclear accumulation during inflammation that still need to be addressed in future studies. Recent work suggests that several mitochondrial enzymes can translocate into the nucleus and constitute a form of mitochondria-to-nucleus communication[29] and that mitochondria-derived vesicles or chaperone proteins and nuclear transcription factors may promote the entry of selected mitochondrial proteins into the nucleus via the nuclear pores[33,44]. It is possible that the Mfn2 interaction with β-catenin or other proteins may facilitate the entry via nuclear pores during inflammation but this will need to be addressed in future studies targeting nuclear transport mechanisms. Although Mfn2 lacks a nuclear localization sequence (NLS), we found that its binding to the adherens junction protein β-catenin, which is known to translocate to the nucleus where it acts as a transcriptional co-regulator[30], was increased following inflammatory activation. Sequence alignment analysis indicated that Mfn2 has a putative β-catenin binding motif such as "SxxSSLSxLS" or "DxθθxΦx$_{2-7}$E"

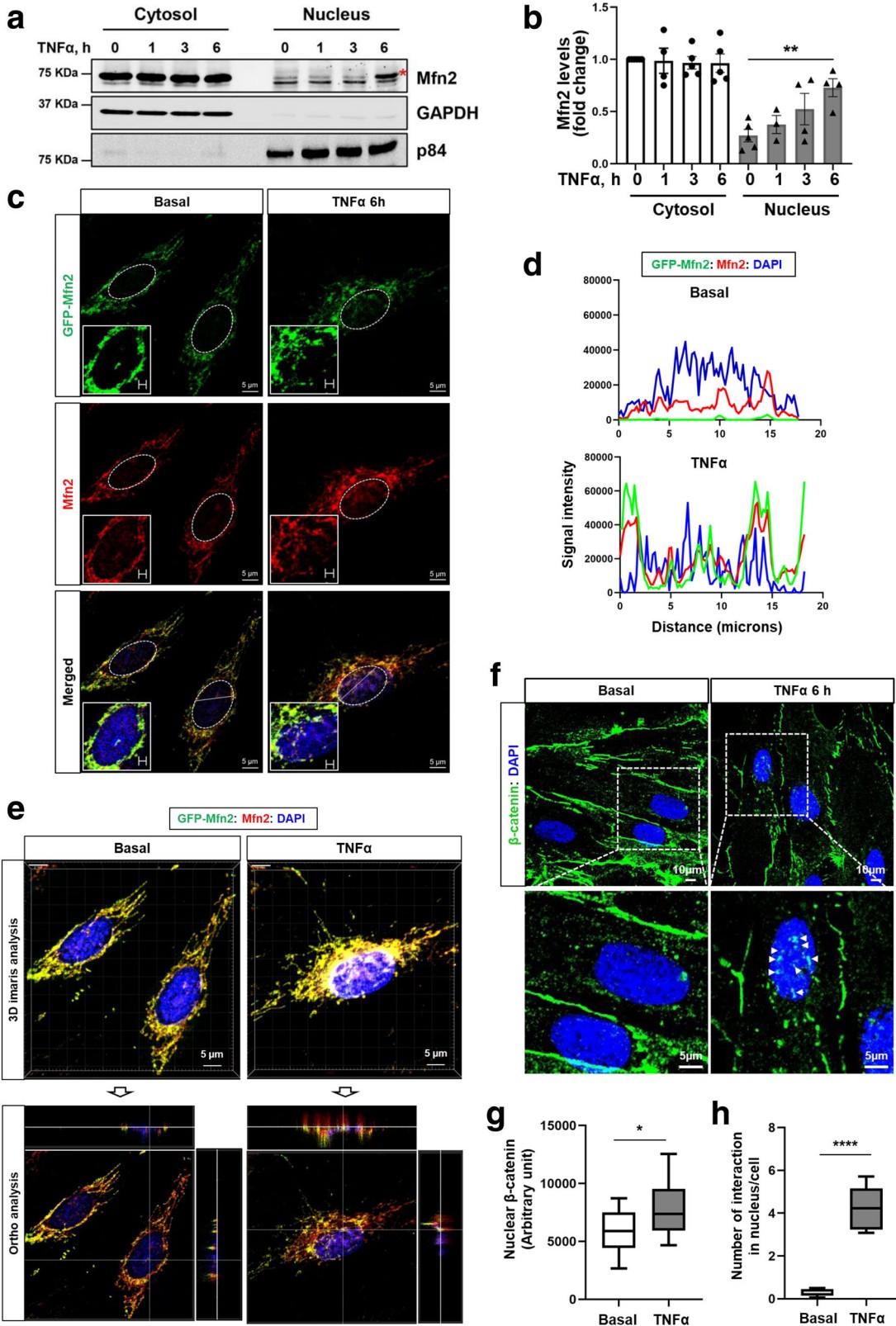

motifs (θ and Φ are hydrophobic and aromatic residues, respectively) between its heptad repeat (HR) domains[45].

The nuclear translocation of β-catenin is essential for canonical Wnt signaling pathway because β-catenin binds DNA and acts as a transcriptional regulator during embryonic development and adult tissue homeostasis[30]. It has been shown that β-catenin translocates into the nucleus despite the lack of an NLS although the underlying mechanisms are not well understood[30]. It is possible that the nuclear accumulation of Mfn2 utilizes a similar mechanism because we found that Mfn2 remains bound to β-catenin when they both disassociate from adherens junctions. This raised the intriguing question whether Mfn2 could

**Fig. 7 TNFα triggers translocation of Mfn2 into the nucleus.** HLMVECs were stimulated with TNFα (10 ng/mL) for the indicated times (0, 1, 3, or 6 h). **a** The cellular localization of Mfn2 was examined by a subcellular fractionation assay. The same amount of cytosolic and nuclear fraction was loaded in the SDS-PAGE gel. GAPDH and p84 were used as controls for cytosolic or nuclear fraction, respectively. The images are representative of at least 3 independent experiments. A red asterisk (*) indicates the nuclei translocated Mfn2. Uncropped blots are provided in the Source Data file. **b** Quantification of Mfn2 protein levels in (**a**) with ImageJ. Data are mean values ± SEM for 3–5 independent experiments. **$p = 0.0054$ for TNFα 0 h vs 6 h in nucleus by paired, two-tailed t-test. **c** HLMVECs expressing GFP-Mfn2 were stimulated with TNFα (10 ng/mL) for 6 h and fixed with 100% cold (−20 °C) methanol on ice for 5 min and permeabilized with 0.25% Triton X-100 for 10 min. Cellular localization of exogenous GFP-Mfn2 was determined by GFP and endogenous Mfn2 was examined by Mfn2 immunostaining. DAPI (blue) indicates nuclei. The images were obtained by Z sections (>87 sections) with x2.4 zoom using confocal microscopy (Zeiss LSM770, Plan Apo 1.46NA, ×63 magnification) and represented by one section. The white boxes are the enlarged nucleus and white lines in merged images indicate the area where the distances between GFP-Mfn2, Mfn2, and DAPI were analyzed using ImageJ (**d**). **e** 3D image analysis with Imaris software in (**c**) and ortho analysis to show co-localization of GFP-Mfn2, Mfn2, and DAPI. The images are representative of at least three independent experiments. **f** HLMVECs stimulated with TNFα (10 ng/mL) were fixed with the same method in (**c**), and β-catenin cellular localization was examined by β-catenin immunostaining and the images were taken using confocal microscopy (Zeiss LSM880, Plan 1.46NA, ×63 magnification). **g** The fluorescence intensity of β-catenin in nuclei in immunostained images of (**f**) was quantified using ImageJ. Data are presented with box and whiskers plot and whiskers are Min to Max. $n = 27$–31 biologically independent cells. *$p = 0.0155$ by unpaired, two-tailed t-test. **h** Interaction of endogenous Mfn2 and β-catenin in HLMVECs was examined with a mouse Mfn2 antibody and rabbit β-catenin antibody with or without TNFα (10 ng/mL) for 6 h by proximity ligation assay. Quantification of number of interactions within the nucleus per cell was determined by counting the number of dots by interacting of Mfn2 and β-catenin. Data are presented with box and whiskers plot and whiskers are Min to Max. $n = 8$–13 biologically independent samples. ****$p < 0.0001$ by unpaired, two-tailed t-test.

functionally regulate gene expression, possibly by affecting the transcriptional activity of β-catenin. We found that nuclear Mfn2 indeed inhibits β-catenin transcriptional activity for key pro-inflammatory cytokines such as *IL-6* or *IL-18* as well as an adhesion molecule *ICAM-1* during inflammation. Interestingly, β-catenin depleted ECs demonstrated significant decreases in the gene expression of selected pro-inflammatory genes such as *ICAM-1*, *IL-1β*, or *IL-18*, but β-catenin depletion did not impact other pro-inflammatory genes such as *IL-6*, *caspase1*, or *VCAM-1*, indicating a specificity of the inflammatory genes regulated by β-catenin. A non-canonical role for transcriptional regulators in endothelial barrier regulation has also recently been reported for Notch signaling[46]. Possible future studies could address whether Mfn2 also modulates the role of Notch in regulating the endo-thelial barrier.

Although mitochondria contain an independent genome, the vast majority of mitochondrial proteins are encoded in the nuclear genome[32]. However, less is known about how mito-chondria engage in retrograde communication with the nucleus and could potentially modulate nuclear gene expression. Even though it came as a surprise that Mfn2 accumulated in the nucleus, recent studies are increasingly finding mitochondrial proteins in nuclei. Multiple proteins that were previously thought to be exclusively mitochondrial have recently been shown to also localize within nuclei, yet their nuclear function is not fully understood[32,33,44].

Importantly, we demonstrate here that Mfn2 and β-catenin are both necessary for EC barrier integrity in vitro and in vivo. Genetic in vivo deletion of Mfn2 specifically in the endothelium resulted in a disruption of vascular homeostasis, as evidenced by increased lung vascular permeability and an increase in inflam-matory gene expression along with increasing inflammatory immune cells even in the absence of any additional pathogenic stimuli. The lung endothelium may be especially vulnerable to inflammatory activation as a recent comparative analysis of vas-cular endothelial cells in distinct organs found that the lung endothelium had the highest expression of inflammatory and immune response genes even during homeostasis when compared to the endothelium of other organs such as the brain or the heart[47]. The association between Mfn2 and vascular permeability may thus represent promising therapeutic targets, to reduce the influx of inflammatory cells[3,48] as well as to reduce tumor metastasis or tumor growth[49]. A recent study also identified an important link between angiogenesis and endothelial barrier

maturation by demonstrating that the transcriptional regulator YAP/TAZ was involved in both processes[41]. Our study estab-lished the barrier stabilization role for Mfn2 in the endothelium, but it is possible that Mfn2 also may regulate angiogenesis.

In summary, our findings reveal intriguing non-mitochondrial roles for Mfn2 as a stabilizer of the vascular barrier and an endogenous suppressor of inflammatory gene expression. These insights could help us better understand the role of "mitochon-drial" proteins in the nucleus and pave the way for future anti-inflammatory therapies.

## Methods

**Reagents.** CM-H$_2$DCFDA [5-(and-6)-chloromethyl-2′,7′-dichlorodihydro fluor-escein diacetate, acetyl ester, Invitrogen C6827] was obtained from Molecular Probes. DCP-Bio1 was obtained from EMD Millipore (Kerafast, NS12265MG). Catalase (#219008) was obtained from Calbiochem (USA). Other reagents were obtained from Sigma.

**si/shRNA or plasmid construct.** Lenti-GFP-Mfn2 or lenti-shMfn2 RNA con-structs were generated in our laboratory. shMfn2 RNA oligos were inserted into a (sense 5′- CCG GGC TCA GTG CTT CAT CCC ATT TCT CGA GAA ATG GGA TGA AGC ACT GAG CTT TTT -3′, anti-sense 3′- AATTAA AAA GCT CAG TGC TTC ATC CCA TTT CTC GAG AAA TGG GAT GAA GCA CTG AGC -5′, target of 3′UTR from 3237-3257) lentiviral Tet-pLKO-puro backbone vector (Addgene, #21915). GFP-Mfn2 was subcloned into a mammalian lentiviral pWPXL expression vector (Addgene, #12257). A β-catenin reporter (M50 super 8x TOP-flash, Addgene, #12456) was provided by Dr. Kishore K Wary at University of Illinois at Chicago. pRL/TK (renilla-luciferase) was provided by Dr. Chinnaswamy Tiruppathi at University of Illinois at Chicago. pRRL-lifeAct-GFP was provided by Dr. Peter Carmeliet at the Center for Cancer Biology, VIB, Leuven, Belgium. We obtained siRNA for beta-catenin (sc-29209), Mfn2 (sc-43928), Mfn1 (sc-43927), Drp1 (sc-43732), Opa1(sc-106808) or control (sc-37007) from Santa Cruz Biotechnologies.

**Lentivirus production and purification.** HEK293T (CRL-11268, ATCC) cells were used to produce lentivirus and the cells were transfected with DNAs (2.5 μg pMD2. G, 5 μg of psPAX2, and 7.5 μg of DNA expression vector) with 30 μg poly-ethylenimine (PEI, Polysciences, 23966, USA) in DMEM media containing 10% FBS without antibiotics overnight and changed with fresh DMEM media supple-mented with 10% FBS and 1% Pen/Strep and then incubated for 48 h at 37 °C. The media containing secreted lentivirus was collected and the virus was purified using Lenti-X concentrator (CloneTech, 631232).

**Cell culture.** Human lung microvascular endothelial cells (HLMVECs, CC-2527, Lonza) were obtained from Lonza and cultured with EGM2 (Lonza) including all supplements and 10% FBS (Hyclone) until passage 8. The cells were transduced with doxycycline inducible lentiviral shMfn2 RNA with 1:2000 dilution polybrene (Millipore, TR-1003-G) for 24 h after which the media was changed. After 48 h, the cells were treated with doxycycline (200 ng/mL) or DMSO (control) for 72 h to

knockdown endogenous Mfn2. As a control, we used lenti-Mfn2 shRNA expressing cells which were treated with DMSO.

**Three-dimensional structured illumination microscopy (3D-SIM).** To visualize Mfn2 in AJs proteins complex, we used the recently developed technology of 3D-SIM which provides high spatial resolution[18,19]. HLMVECs were cultured on #1.5 coverslips (~170 μm thickness) with restricted thickness-related tolerance (±5 μm) (MatTek: high tolerance coverslips, pcs-170-1818). Confluent ECs were fixed with 100% cold (−20 °C) methanol for 5 min on ice or with 4% paraformaldehyde (PFA) for 10 min at room temperature. The fixed cells were permeabilized with 0.25% Triton X-100 at room temperature for 10 min and then blocked with blocking buffer (1X PBS, 2% BSA, and 0.05% Tween 20) for 1 h followed by immunostaining with 1:250 dilution of anti-Mfn2 (ab56889, mouse; 12186-1-AP, rabbit), anti-Tom20 (sc-14415, rabbit; sc-17764, mouse), or anti-Mfn1 (CST-14739, rabbit) overnight at 4 °C and then stained with 1:500 of secondary antibodies (mouse Alexa488, mouse Alexa568, rabbit Alexa488, or rabbit Alexa568) for 2 h at room temperature (RT). After secondary antibodies for Mfn2 or Tom 20, the VE-cadherin was immunostained with 1:250 dilution of Alexa Fluor647 mouse anti-human CD144 (BD561567) for 1 h at room temperature. The nuclei were visualized by separately staining with DAPI. After mounting with Prolong gold antifade reagent (Invitrogen, P36934) without DAPI, the images were taken at room temperature by using 3D-SIM (DeltaVision OMX SR, GE Healthcare, Life Sciences) equipped with an Olympus 60X/1.42 NA Plan Apo objective and refractive index matched immersion oil, n = 1.516-1.518. Z-axis sections (125 nm step size) were taken at full-frame structured illumination mode (1024 × 1024 pixel, sequential acquisition). Softworx (Applied Precision) was used to reconstruct 3D-SIM images. The projection images are presented and the protein co-localization was analyzed with plot profiles and Manders' overlap coefficient using ImageJ (NIH, USA).

**Immunofluorescence imaging using confocal microscopy.** To evaluate EC barrier integrity, confluent HLMVECs in 6 well plates containing coverslips were treated with or without TNFα (10 ng/mL) for 6 h and fixed with 4% PFA for 10 min at RT and then blocked with blocking buffer (1X PBS, 2% BSA, and 0.05% Tween 20) without permeabilization for 1 h. To evaluate co-localization of Mfn2, Tom20 and AJs proteins, the cells were treated with or without TNFα (10 ng/mL) for 6 h and fixed with 4% PFA for 10 min at RT. The fixed cells were permeabilized with 0.25% Triton X-100 at room temperature for 10 min and then blocked with blocking buffer (1X PBS, 2% BSA, and 0.05% Tween 20) for 1 h. The primary antibodies (VE-cadherin; Cayman #160840, β-catenin; ab32572, SC-7963, or Mfn2; 12186-1-AP, Mfn1; CST-14739 or Tom20; SC-17764) were diluted 1:250 with antibody dilution solution (1X PBS, 1% BSA, and 0.05% Tween 20) overnight at 4 °C and then incubated with 1:500 dilution of secondary antibody (Goat anti-Rabbit or anti-mouse IgG (H + L) Highly Cross-Adsorbed Secondary Antibody, Alexa488 or Alexa546 for 1 h at room temperature. To evaluate co-localization of exogenous GFP-Mfn2 and endogenous Tom20, HLMVECs expressing GFP-Mfn2 were immunostained with Tom20 specific antibody (sc-14415, rabbit, 1:250 dilution) and anti-rabbit secondary Alexa633 (1:500 dilution). The nuclei were visualized by separately staining with DAPI or Hoechst. After mounting with Prolong gold antifade reagent without DAPI, the images were taken using confocal microscopy (Zeiss LSM880, Plan Apo 1.46NA, 63x objective). To verify Mfn2 nuclear localization, the samples were Z sectioned (average 91 slices) with equal fluorescence intensity and the images were rendered in 3D with interpolation using Imaris x64 version 9.5.0 (Bitplane Scientific). Mfn2 nuclear localization was evaluated by the co-localization of GFP-Mfn2, Mfn2, and DAPI in section # 52 by ortho analysis (ZEN blue). For rescue experiments with GFP-Mfn2, Mfn2-KD ECs were transduced with lentiviral GFP-Mfn2. The control (DMSO treated cells), Mfn2-KD ECs (doxycycline-treated cells), and Mfn2-KD + GFP-Mfn2 OE (doxycycline-treated cells overexpressing GFP-Mfn2) were examined for EC junctional integrity by immunostaining with VE-cadherin or β-catenin antibodies. The images were taken using confocal microscopy (Zeiss LSM770, Plan Apo 1.46NA, 63x objective). Area and fluorescence intensity of VE-cadherin or β-catenin at AJ junctions were analyzed using ImageJ. Full microscopy images are provided in the Source data file.

**Immunoprecipitation assay and western blotting.** Confluent HLMVECs in 100 mm dishes were lysed with immunoprecipitation buffer (50 mM HEPES pH7.5, 120 mM NaCl, 5 mM EDTA, 10 mM Na pyrophosphate, 50 mM NaF, 1 mM Na$_3$VO$_4$, 1% Triton X-100). For TNFα treatment, confluent HLMVECs in 100 mm dishes were treated with or without TNFα (10 ng/mL) for 6 h. For H$_2$O$_2$ treatment, HLMVECs were stimulated with H$_2$O$_2$ (500 μM) for 30 min. The cell lysate used for immunoprecipitation with 1 μg Mfn2 (ab56889) antibody followed by Western blotting with 1:1000 dilution of VE-cadherin (Cayman #160840), β-catenin (ab32572), or Mfn2 (12186-1-AP) antibodies. For non-reducing SDS-PAGE, protein samples were prepared with 1x SDS-gel same buffer without DTT or β-Mercaptoethanol and boiling. Mouse normal IgG (1 μg) was used as a negative control for immunoprecipitation assay. Western blots were performed as previously described[41]. Quantitative analysis of Western blotting was performed using Image J. All raw gel blots are available in the Source data file.

**Albumin tracer permeability assay.** HLMVECs transduced with doxycycline inducible lentiviral Mfn2 shRNA were treated with doxycycline (200 ng/mL) for 72 h. For rescue experiments, Mfn2-KD ECs were transduced with lentiviral GFP-Mfn2 which is not targeted by Mfn2 shRNA because Mfn2 shRNA targets Mfn2 3′-UTR regions. The control (DMSO treated cells), Mfn2-KD ECs (doxycycline-treated cells), and Mfn2-KD + GFP-Mfn2 OE (doxycycline-treated cells overexpressing GFP-Mfn2) were plated on Transwell inserts (Costa CLS3413-48A, 0.4 μm pore size) in a 24 well plate. A confluent monolayer of ECs was treated with FITC-albumin tracer in medium (0.5 mg/mL)[50] in the upper chamber. The infiltrated tracer concentration was measured in the lower chamber media for indicated times at A485 nm (excitation) and A535 nm (emission).

**Transendothelial resistance (TER).** 40,000 cells of control or Mfn2-KD ECs were plated on a gold microelectrode (8W1E + PET, ECIS Cultureware) for TER 1 day before measurement. 250 μL fresh media was added to the cells and the resistance of the EC barrier was measured with an ECIS®-1600 R system (Applied Biophysics, Troy, NY, USA) in accordance with the manufacturer's instructions[22].

**In situ proximity ligation assay (PLA).** Confluent HLMVECs in 6 well plates including glass coverslips were treated with or without TNFα (10 ng/mL) for 6 h. The interaction between VE-cadherin (SC-9989, mouse) and β-catenin (ab32572, rabbit), or between Mfn2 (ab56889, mouse) and β-catenin (ab32572, rabbit) was determined with1:200 dilution by PLA (the Duolink In situ Red Starter Kit, UO92101, Sigma)[25], following manufacturer's instructions with minor modifications.

**Evaluation for total ROS.** HLMVECs were stimulated with TNFα (10 ng/mL) for the indicated time (0, 15 min, 30 min, 3 h or 6 h) and then incubated with 20 μM CM-H2DCFDA (Invitrogen, C6827) for 6 min at 37 °C[41]. DCF fluorescence was measured by confocal microscopy (Zeiss LSM880) with ×63 magnification using the same exposure conditions in each experiment. Relative DCF fluorescence was measured by ImageJ and presented as fold change.

**DCP-Bio1 assay.** To determine sulfenic acid (Cys-OH) of target protein, we used an innovative cell permeable biotin-labeled Cys-OH trapping probe (DCP-Bio1)[29]. Briefly, TNFα stimulated-control or Mfn2-KD ECs were lysed with degassed-specific lysis buffer (50 mM HEPES, pH7.0, 5 mM EDTA, 50 mM NaCl, 50 mM NaF, 1 mM Na$_3$VO$_4$, 10 mM sodium pyrophosphate, 5 mM IAA, 100 μM DPTA, 1% Triton-X-100, protease inhibitor, 50 unit catalase, 200 μM DCP-Bio1), and then pull-downed with streptavidin beads overnight. All steps were performed in the dark. DCP-Bio1 conjugated sulfenylated-proteins were measured with specific antibodies (1:1000 dilution) for Mfn2, β-catenin, or actin using Western blotting.

**Subcellular fractionation.** Confluent HLMVECs in 100 mm dishes were stimulated with TNFα for the indicated times (0, 1, 3, 6 h) followed by subcellular fractionation[51]. Briefly, the cells were lysed with 500 μL lysis buffer (10 mM HEPES, pH7.9, 10 mM KCl, 0.1 mM EDTA, 0.1 mM EGTA, and EDTA free protease inhibitor cocktail), collected in Eppendorf tubes, and incubated in ice for 10 min. The lysate was freshly added to 60 μL 10% NP-40 and then vortexed for 10 s followed by centrifugation at 16,000g for 1 min at 4 °C. The supernatant (cytosolic fraction) was carefully transferred to a second tube. The pellet was washed once with lysis buffer and 200 μL nuclear extract buffer (25 mM HEPES, pH7.9, 0.4 M NaCl, 0.5 mM EDTA, 0.5 mM EGTA, and EDTA free protease inhibitor cocktail) was added by tapping gently and incubated in ice for 30 min with vortexing every 5 min. The nuclear fraction was collected by centrifugation at 16,000g for 5 min at 4 °C. The same amount of cytosol and nuclear fraction was loaded in a SDS-PAGE gel followed by Western blotting. GAPDH (1;1000 dil, 10494-1-AP, rabbit) and p84 (nuclear matrix protein, 1:1000dil, sc-514123, mouse) were used as controls for the cytosolic or nuclear fraction, respectively.

**β-catenin reporter luciferase assay.** Control and Mfn2-KD ECs were transfected with 1 μg of a β-catenin reporter (M50 super 8x TOPflash containing TCF/LEF sites upstream of a luciferase reporter)[52] and 35 ng of pRL/TK using PEI transfection reagent (Polyethylenimine). At 48 h after transfection, the cells were stimulated with TNFα (10 ng/mL) for 6 h and then 100 μL of cell lysate from each sample was used to measure reporter gene expression. Firefly and Renilla luciferase activity were determined by the dual luciferase reagent assay system (Promega). The relative luciferase activity represents the mean value of the firefly/Renilla luciferase.

**Quantitative real-time PCR.** Total RNA was isolated by using phenol/chloroform and TriZol Reagent (Invitrogen, 15596026) as described[41]. Reverse transcription was carried out using high capacity cDNA reverse transcription kit (Applied Biosystems, 4368814) using 2 μg of total RNA. Quantitative PCR was performed with fast start universal SYBR Green master (ROX) PCR kit (Roche, 04913914001) using QuantStudio7 (Thermofisher). Samples were all run in triplicates to reduce variability. Expression of human genes of *Mfn2, ICAM-1, IL-6, IL-18, or IL-1β* was determined using the following primers; **Mfn2**, sense 5′-CATCCCCAGTTGTCCT

CAAG -3′, anti-sense 5′-CAAGCCGTCTATCATGTCCTG-3′, **ICAM-1**, sense 5′-C GTGCCGCACTGAACTGGAC-3′, anti-sense 5′-CCTCACACTTCACTGTCAC CT-3′, **IL-18**, sense 5′-CAGACCTTCCAGATCGCTTC-3′, anti-sense 5′ -GGGT GCATTATCTCTACAGTCAGAA-3′, **IL-1β**, sense 5′-CCAGGGACAGGATATG GAGCA-3′, anti-sense 5′- TTCAACACGCAGGACAGGTACAG-3′, **IL-6**, sense 5′-TAGCCGCCCCACACAGACAG-3′, anti-sense 5′-GGCTGGCATTTGTGGTT GGG-3′. Expression of mouse genes of *Mfn2, IL-1β, IL-6, TNFα or IFNγ* was determined using the following primers; **Mfn2**, sense 5′-CAAGACCGGCTGAGG TTTATT-3′, anti-sense 5′-CCTTTCCACTTCCTCCGTAATC-3′, **IL-1β**, sense 5′-CCTTCCAGGATGAGGACATGA-3′, anti-sense 5′-TGAGTCACAG AGGATGGGCTC-3′, **IL-6**, sense 5′-CTTCCATCCAGTTGCCTTCTTG-3′, anti-sense 5′- AATTAAGCCTCCGACTTGTGAAG-3′, **TNFα**, sense 5′-ACGGCATG-GATCTCAAAGAC-3′, anti-sense 5′- AGATAGCAAATCGGCTGACG-3′, **IFNγ**, sense 5′-ACAATGAACGCTACACACTGCAT -3′, anti-sense 5′-TGGCAGTAAC AGCCAGAAACA-3′. Human primers for **β-catenin** (QT00077882) or mouse primers for **HPRT** (QT00166768) were purchased from Qiagen. Expression of genes was normalized and expressed as fold-changes relative to **B2M** (internal control; sense 5′-GGTTTCATCCATCCGACATT-3′, anti-sense 5′-ATCTTTTTC AGTGGGGGTGA-3′) or to HPRT. A complete list of all primers is available in Supplementary Table 1 of the Source Data files.

**Animal experiments.** All animal studies were carried out following protocols approved by the Animal Care and Institutional Biosafety Committee of the University of Illinois at Chicago. Mice of both genders at 8-10 weeks were used because gender-specific effects could not be recognized. Mice were maintained under standard conditions (standard diet and water) at 23 ℃ and ~60% humidity with 12 h light and 12 h dark cycles. We purchased Mfn2 floxed mice from Jackson Laboratory (# 026525). Mfn2 flox/flox female mice were crossed with tamoxifen-inducible VE-Cad-Cre transgenic male mice[53] (Cdh5^ERT2, tamoxifen-inducible endothelial-specific Cre recombinase) to derive inducible EC-specific conditional Mfn2 knockout (Mfn2^{EC−/−}) mice. The mice aged 8-10 weeks received tamoxifen (2 mg/100 μL in corn oil, 20-25 g body weight, i.p.) for 3 consecutive days to induce Cre-mediated recombination and were then rested for 1 month prior to the experiments.

**Evans blue albumin (EBA) assay.** Mfn2^{fl/fl} (littermate control) and Mfn2^{EC−/−} mice were anesthetized and Evans blue-albumin (EBA) (40 mg/kg) was injected into the right jugular vein and allowed to circulate in the blood vessels for 45 min. Mouse lungs were excised, weighed, homogenized in 1 ml PBS, and extracted in 2 ml formamide for 24 h at 60 ℃. Evans blue content in the formamide extract was determined by $OD_{620}$ and $OD_{740}$. We used the following equation:[54] corrected 620 nm optical density = $(OD_{620}$ (Evans blue) $- (1.426 \times OD_{740}$ (hemoglobin) $+ 0.03)$.

**Lung endothelial cell isolation.** The lungs of anesthetized mice were perfused with PBS and the lungs were dissected and placed in a 50 mL tube with PBS. The lung tissues were minced as small as possible with scissors. 3 mL collagenase type 1 (2 mg/mL) was added and the tissues were incubated for 30 min at 37 ℃ with gentle shaking[3]. At the end of the digestion process, the tissue was titrated using 18 G needles in syringes up and down 5 times and the cell suspension was filtered through 40 μm disposable cell strainer into a fresh 50 mL tube. The cells were then washed twice with isolation buffer (PBS + 0.5% BSA + 2 mM EDTA + 4.5 mg/mL D-glucose). The endothelial cells were incubated with 5 μg/mL rat anti-mouse-PECAM-1 (CD31) antibody (BD Pharminogen 553370) for 30 min at 4 ℃ and isolated with magnetic beads sorting using Dynabeads (sheep anti-Rat IgG, #11035, Invitrogen)[49]. Finally, CD31 positive ECs were used for Western blotting.

**Acute lung inflammation.** C57/BL6 mice received sublethal LPS 8 mg/kg i.p. (Sigma #L2630)[3] or PBS as control and sacrificed after 6 h or 24 h. Whole lungs were excised and homogenized, and protein was extracted for the DCP-Bio1 assay.

**Lung histology and FACS analysis for lung inflammation.** Briefly, 4% paraformalin-fixed and 5 μm paraffin-embedded sections of lung tissues were used for H&E[23]. Images were taken using a Revolve fluorescence microscope (Echo). Single cell suspensions were isolated from the whole lung tissue of Mfn2^{fl/fl} and Mfn2^{EC−/−} mice in homeostatic conditions[23], and stained with 1:100 dilution of the indicated antibodies: anti-CD45 (A20), anti-CD11c (N418), anti-Siglec F (S17007L), anti-F4/80 (BM8), anti-CD3 (17A2), anti-B220 (RA3-6B2), anti-Ly6G (1A8) and anti-NK1.1 (PK136) purchased from BioLegend. 1) myeloid cells: interstitial macrophage 2 (IM2), dendritic cells (DC) with CD11c and F4/80; Neutrophils, monocyte (Mono), and interstitial macrophage 1(IM1) with F4/80 and Ly6G, 2) Lymphoid: B cell and T cells with CD3 and B220; NKT cells and T cells with CD3 and NK1.1. Samples were run through a Gallios flow cytometer (Beckman Coulter, Pasadena, CA) and analyzed by Kaluza software (Beckman Coulter). The inflammatory response cells were represented by the percent (%) of whole lung cells.

**Statistics and reproducibility.** Quantitative analysis of Western Blotting was performed using ImageJ (NIH) software. Co-localization of fluorescence images by confocal microscopy were analyzed by evaluating Manders' coefficiency using ImageJ. Quantification is presented as the mean ± SEM from at least 3 independent biological replicate experiments. Data distribution was analyzed with the Shapiro-Wilk test[55] to assess for normal distribution. For normally distributed data, the Student's $t$ test, one-way and two-way ANOVA with Bonferroni post-tests were used for between group comparisons to determine statistical significance, with a $p$ value threshold of less than 0.05. When data were not normally distributed, we used the non-parametric Mann-Whitney test for between group comparisons. Significance levels are indicated in the figures as $*p < 0.05$, $**p < 0.01$, $***p < 0.001$, and $****p < 0.0001$. All statistical analyses were conducted using Prism 9, GraphPad Software (La Jolla, CA).

**Reporting summary.** Further information on research design is available in the Nature Research Reporting Summary linked to this article.

## Data availability

The authors declare that all data supporting the findings of this study are available within the paper and its supplementary information files. Source data files provided with this paper include proteomic datasets and microscopy images. The mass spectrometry proteomics data have been deposited to the ProteomeXchange Consortium via the PRIDE partner repository with the dataset identifier PXD024620 and 10.6019/ PXD024620. Source data are provided with this paper.

## Code availability

No new code was generated for this work.

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

## Acknowledgements

The studies were supported by NIH grants P01-HL60678 (JR), T32-HL007829 (to SK and AJ), T32-HL139439 (to SK), R01-HL154538 (to JR), R01-HL126516 (to JR), R01-HL152515 (to JR), AHA CDA grant 19CDA34680000 (to YMK), and R01-HL126852 (to GYP). The Cdh5-CreERT2 mice were kindly provided by Dr. Ralf Adams. This research is based in part upon work conducted using the UNC Proteomics Core Facility, which is supported in part by P30 CA016086 Cancer Center Core Support Grant to the UNC Lineberger Comprehensive Cancer Center. We were assisted by the Fluorescence Imaging Core at the Research Resources Center of the University of Illinois at Chicago.

## Author contributions

Y.M.K. and J.R. developed the overall study design, Y.M.K., S.K. and H.G.M. performed the experiments, Y.M.K., S.K., H.G.M., A.J. and I.S. performed the overall data analysis, P.T.T. performed the image analysis, Y.M.K. and J.R. wrote the initial manuscript draft. G.Y.P. designed the experiments to study immune cell phenotypes. All authors provided critical feedback and revisions for the manuscript.

## Competing interests

The authors declare no competing interests.

## Additional information

17