## [Peer Review File · Nature Communications]

Reviewers' comments:

Reviewer #1 (Remarks to the Author):

The manuscript by Kim et al. describes a novel role of mitochondrial membrane protein Mitofusin 2 (Mfn2) in maintaining vascular barrier integrity and protecting against endothelial inflammation. By employing advanced imaging techniques and varieties of molecular/biochemical assays in both cell culture and animal models, the authors analyzed Mfn2 localization at adherens junctions and interaction with adherens junction protein β -catenin which might be essential for non-canonical function of Mfn2 in endothelium in inflammatory conditions. Inflammatory agonist-induced ROS production leading to the sulfenylation of Mfn2 and its subsequent nuclear translocation along with β -catenin has been postulated as a major mechanism regulating the functional role of Mfn2 in endothelial cells. Although the main hypothesis is original and intriguing, the study requires more substantial evidence of proposed mechanism, clarification of experimental details and data analysis.

Major points:

1. The role of Mfn2 in ER stress, apoptosis and mitophagy has been described in other cell types and same phenomenon may exist in endothelial cells. Normal mitochondrial dynamics is essential for endothelial function, and elevated ROS, Ca^{++} levels resulted from knockdown of Mfn2 may have direct deleterious effects on endothelial cells. ROS formation resulting from Mfn2 knockdown may lead to oxidant stress and inflammatory signaling or further augmentation of LPS-induced inflammation, which is another alternative to the proposed mechanism. This needs to be experimentally tested.
2. Fig-1: selection of Tom20 as mitochondrial marker is sub-optimal. Fluorescence signal intensity is marginal, the mitochondrial arrangement is poorly visible. Other more definitive markers need to be demonstrated.
3. Co-localization of Mfn2 and Tom2 is rare as shown by super-resolution microscopy. How can this be explained?
4. Patterns of IF-staining of VE-cadherin and β -catenin (Fig-1F) and PLA analysis (2C) of same proteins do not correspond, thus raising concern about optimal conditions for PLA assay.
5. Data of surface biotinylation assay are really confusing. β -catenin and Mfn2 are cytosolic proteins, and their biotinylation is a major concern. Presented decrease of biotinylated β -catenin and Mfn2 in TNF-treated cells does not provide any clue to jump to the conclusion that they are being released from assembled junctions.
6. TNF α causes VE-cadherin internalization, which is inconsistent with surface biotinylation data in Fig-1D.
7. Blot reprobing with the antibody used for pulldown is essential as IP control in all co-IP expts.
8. More definitive proof of Mfn2 nuclear localization is needed using 3D image reconstruction, given the reported close positioning of mitochondria and nucleus in certain

conditions.

9. Fig-2A – western blot data not convincing and contradictory. No apparent difference of both, VE-cadherin and b-catenin association with Mfn2 between control and TNF-stimulated conditions. TNF causes dissociation of AJ complex and VE-cadherin internalization, which is not evident from presented data.

10. Fig-4B,D - western blot data are not convincing: protein changes are marginal, problematic to judge.

Minor points:

1. To better establish a barrier protective/anti-inflammatory role of Mfn2, it is essential to perform experiments with Mfn2 overexpression to test if such strategy rescues agonist-caused endothelial dysfunction.

2. The interpretation of Fig. 3C is unclear. How can it be confirmed that the two bands appearing in the absence of reducing agent are disulfide bond-linked Mfn2 and b-Catenin? Further experimental evidence is required.

3. Fig. 1B needs to be mentioned in the text. Similarly, there are some phospho-beta catenin blots in supplementary figs, which also need to be mentioned.

4. Was there any specific reason to use doxycycline-inducible lentiviral shRNA for Mfn2 knockdown? Why regular siRNA transfection cannot be used?

5. Fig. 3F, mice were given 8mg/kg of LPS i.p. This is a sub-optimal model of lung injury.

Reviewer #2 (Remarks to the Author):

The authors report a set of observations related to the role of Mfn2 in adherens junctions and related inflammatory response. This interesting role, previously unreported, is based on several different measurements. This includes mass spec, biochemistry, and microscopy.

I have primarily focused on the microscopy here, since it is my main expertise.

1. It is important to note that Alexa 647 contains an impurity that fluoresces in the near-red (Stone MB, Veatch SL. Far-red organic fluorophores contain a fluorescent impurity. *Chemphyschem*. 2014;15(11):2240–2246. doi:10.1002/cphc.201402002). So, the authors' use of Alexa647 together with Alexa595 to look for co-localization (Fig. 1A) is unfortunate, since it may be influenced by this impurity. This is underlined here, where the “colocalization” signal in the 595 channel is very weak (low SNR). To demonstrate co-localization, the authors should use additional pairings of labels, such as using Mfn2-GFP and a near-red dye for mitochondria together with VE-cadherin Alexa647, and the pair of Mfn2-GFP with VE-cadherin Alexa647 alone.

2. The data referenced to show that Mfn2 does not necessarily colocalize with mitochondria (SFig 2A) are not convincing. Under inflammatory conditions, there is still a colocalization, which becomes more obvious when increasing the gain. Also a quantification for

colocalization is missing.

3. The authors cite the expected resolution achievable by SIM of 130 nm, but they do not seem to measure it for their images. A quick check shows that at least for some of the data included the pixel-based resolution is quite low, ~300 nm. However, the claimed resolution may not be necessary for the conclusions drawn.
4. Overall, the staining of mitochondria is of low SNR, to the extent that it is sometimes difficult to recognize individual mitochondria (Figure 1A).
5. The image thresholds appear to have been changed, resulting in intensity saturation (enlargements, Fig. 1B).
6. Scale bars seem to be wrong in Panels 1B, 1C, 2C, 4F (not consistent between different magnifications), missing in 2E.
7. Figure 1A and 1C show identical panels
8. The disruption of adherens junctions upon Mfn2 knockdown is striking (Fig. 1F), but should be quantified.
9. Differences upon TNF-alpha treatment appear convincing (2E). However, the quantification of differences does not appear to have been done on a sufficient sample size: Fig 2D and F each show only 6 data points each, not enough to know how the data is distributed.
10. For some datasets, I could not find a description of how the analysis was performed (for example Fig. 2D. F "number of interaction").
11. For statistics, the authors described that they have used student's t-test for comparisons. However, if the data are not normally distributed, a non-parametric test has to be used. For example, data in Figures 2D, 2F, 3B, 3C, 4G should be checked for whether they are normally distributed.

Reviewer #3 (Remarks to the Author):

NCOMMS-19-32607 Review

In this manuscript, the authors investigate the role of mitofusin2 (localised to the outer membrane of mitochondria) in adherent junction complexes (localised at the plasma membrane), focussing on their importance in endothelial barrier and transcriptional responses upon inflammatory stimulation. They convincingly demonstrate (in cells and in mice lungs) a totally novel role of Mfn2 (or at least of mitochondrial morphology) in restricting endothelial permeability. Consequently, Mfn2 prevents beta-catenin-dependent induction of pro-inflammatory genes, proposed to occur via oxidative-dependent modification of beta-catenin. They observe that Mfn2 locates proximal to the plasma membrane, via association with VE-cadherin under basal conditions. Upon TNFalpha stimulation, beta-catenin dissociates from VE-cadherin and further associates with Mfn2. This is a very interesting novel concept and the title is a fair assessment of their findings.

However, some main conclusions are not fully supported by the data presented. First, the authors imply a non-mitochondrial role of Mfn2, instead depending on a cytoplasmic and nuclear translocation of soluble Mfn2. In fact, Mfn2 is a transmembrane protein and thus it is difficult to envision how the full length protein could be soluble. Therefore, they need to either definitely exclude a role of mitochondria or revise their model. Equally, a direct effect of nuclear (soluble?) Mfn2 on beta-catenin transcription should be either fully addressed or the model should be revised.

Main points:

Mfn2 depletion disrupts AJ junctions. But what is really the role of Mfn2? The authors

propose that Mfn2 interaction with VE-cadherin and beta-catenin in presence/absence of TNF alpha require its re-location outside mitochondria. However, this must be convincingly demonstrated (see technical comments below). Or, instead, could beta-catenin signalling depend on the presence of mitochondria at the plasma membrane, where Mfn2 would act as a tether between both "organelles"? Alternatively, could it instead depend on mitochondrial morphology, rather than on Mfn2 itself? Does it depend on Mfn1 or Opa1? Can it be suppressed by restoring tubulation, e.g. upon depletion of both fusion and fission components?

The nuclear localization of Mfn2 (Fig. 4, see detailed technical comments below) needs to be convincingly demonstrated. Instead, does Mfn2 interact with beta-catenin and with nuclear acid binding proteins in the cytoplasm? Indeed, externalization to the cytoplasm of both nuclear and mitochondrial DNA has been recently shown. An experiment similar to the one presented in Sup1 instead using nuclear/PM/mitochondrial and soluble fractions should allow clarifying this point.

The authors convincingly demonstrate formation of disulphide bonds in beta-catenin upon oxidative stress, which co-precipitate with Mfn2. However, they state without demonstrating that it is an S-S link between Mfn2 and beta-catenin. Moreover, whether beta-catenin S-S bond is relevant or depends on Mfn2 is not analysed.

Further technical / clarity concerns:

They could broaden the references mentioned in the introduction and include more review papers for a non-specialist audience. The reason for specifically studying Mfn2 should be added. A review reference for the pro- or anti-fusion roles of Mfn2 in mitophagy should be included. Reference 19 is controversially discussed and this should be mentioned.

Fig. Sup1: The raw data with the identity and quantification of all Mfn2 interactors depicted in S1B-C found should be included. This is essential to assess the relevance of their further studies. Importantly, Mfn2 is mostly facing the cytoplasm, thus interaction with non-mitochondrial proteins does not allow to exclude that it does so while still located to mitochondria.

Fig. Sup 2A: Absence of co-staining between Tom20 and Mfn2 does not allow excluding Mfn2 from mitochondria. This is a critical point that needs to be convincingly demonstrated. Co-immuno-staining but also biochemical fractionation with other mitochondrial markers (OM, matrix, IM and IMS-located) must be performed.

How specific is the Mfn2 antibody or siRNA? Does it also affect Mfn1?

Fig. 1: Include co-staining between Mfn1, Mfn2 and VE-cadherin, and also between Mfn2, Tom20 VE-cadherin upon TNF alpha. Figure 1B is not mentioned in the text. In 1D include a negative control, e.g. Tom20, Mfn1 or other interacting proteins from the pool-down shown in sup1. In 1F, how does Mfn2-KD change the localization of VE-cadherin and beta-catenin upon TNF alpha, and how does this affect the localization of mitochondria? This is important to interpret the biological significance of the results shown in Fig. 2 C-D and E-F. Also, in 1F, include Mfn2 rescue experiments.

Fig. Sup 2B: include Mfn1 control and Mfn2 rescue experiments.

Lines 150-153: the text does not accurately describe the experiments performed.

Fig. 2A/B Sup 3A: They describe increased Mfn2- beta-catenin interaction upon TNF alpha. However, in the initial coIP from Sup1, they only observe Mfn2- beta-catenin interaction without TNF alpha. Please explain the discrepancy.

Sup3B: How specific to Mfn2 is the GTPase activity observed? Include the value with IgG control.

Fig. 2E: draw also here the boundaries, similar to 2C.

Lines 175-176: it is not necessarily the same Mfn2 population that interacts with VE-cadherin and beta-catenin in presence and absence of TNS alpha. I recommend to tone down.

Fig. 3: It is very clear that Mfn2 - beta-catenin interaction depends on ROS. However, there

is no proof that Mfn2 is forming a S-S link with beta-catenin. Thus, in 3C, perform IB also with Mfn2. Further, analyse beta-catenin S-S formation in Mfn2-KD. Moreover, to prove their claim that interaction is mediated by S-S formation (lines 177-178), they would need to identify the responsible cysteines and observe abolished inflammatory responses in the respective mutants. In the same line, the conclusion presented (lines 199-202) should be proven or toned down.

Fig. 4 and Sup 4B: As stated above, the claimed nuclear translocation of Mfn2 must be convincingly demonstrated or eliminated from the manuscript. After all, their experiments do not address what Mfn2 would do in the nucleus. In 4A,F: Statistical analyses for Mfn2 must be included, other mitochondrial proteins (e.g. Mfn1, Tom20, mitotracker) must be tested. In fact, in 1F, "nuclear" localization of Mfn2 is also visible under basal conditions. In 4B: Simultaneous fractionation of mitochondrial, plasma membrane, nuclear and soluble fractions must be included, analysed by respective markers (including Mfn1, Tom20 and other mitochondrial proteins and also including beta-catenin and other cytosolic-nuclear proteins known to translocate upon TNF alpha. How do they explain that "nuclear" increase is much stronger for Mfn2 than for beta-catenin (Fig. 4B-E)? In fact, the nuclear translocation of beta-catenin is also not very strongly observed by cellular staining, in Sup 4B. It rather appears that it is perinuclearly accumulating in the cytosol. Thus, the interaction of both Mfn2 and Sup 4B with nucleic acids could occur outside the nucleus.

Fig. 5A: For VCAM-1 there is no Mfn2 dependence and it is not mentioned in the text. It could be eliminated. Fig. 4E: include the analysis of simultaneous KD of Mfn2 and beta-catenin.

There are several typos throughout the text and lost numbers within the sup figures. Please correct.

Point by Point Response to reviewers for manuscript NCOMMS-19-32607

Reviewer #1 (Remarks to the Author):

“The main hypothesis is original and intriguing, but the study requires more substantial evidence of proposed mechanism, clarification of experimental details and data analysis.”

Major points:

1. The role of Mfn2 in ER stress, apoptosis and mitophagy has been described in other cell types and same phenomenon may exist in endothelial cells. Normal mitochondrial dynamics is essential for endothelial function, and elevated ROS, Ca⁺⁺ levels resulted from knockdown of Mfn2 may have direct deleterious effects on endothelial cells. ROS formation resulting from Mfn2 knockdown may lead to oxidant stress and inflammatory signaling or further augmentation of LPS-induced inflammation, which is another alternative to the proposed mechanism. This needs to be experimentally tested.

We have now addressed this question whether Mfn2 depletion could have wide-spread effects on cell stress. We found that Mfn2 depletion did not affect ER stress, apoptosis, and mitophagy during homeostatic conditions. Moreover, Mfn2 depleted ECs did not have increased mitochondrial ROS production even though the Mfn2 depletion was sufficient to impair its canonical function of mitochondrial fusion (Fusion impairment shown in the “**Reviewer only**” **Figure1**). These additional data strongly support the notion that the observed effects Mfn2 depletion on the loss of endothelial barrier integrity is a specific finding and does not reflect generalized cell stresses. We present these additional data on the absence of cell stress during homeostatic conditions in the revised manuscript (**new Supplementary figure 3**).

2. Fig-1: selection of Tom20 as mitochondrial marker is sup-optimal. Fluorescence signal intensity is marginal, the mitochondrial arrangement is poorly visible. Other more definitive markers need to be demonstrated.

We agree with the reviewer that the co-localization of Mfn2 and the mitochondrial protein Tom20 can be better visualized and demonstrated than in the images we had originally shown. Concerns about potential imaging “impurities” have been addressed by using antibodies with other wave lengths that avoid potential imaging artifacts. In response, we have therefore performed these studies using confocal and super-resolution microscopy and thus increased the robustness of the findings. We can now demonstrate a much better visualization of mitochondrial structure by staining for Mfn2, Tom20, and Mfn1 staining. We also analyzed protein co-localization using ImageJ and included Manders’ overlap or Spearman’s correlation coefficients. The new data in the revised manuscript clearly address these concerns. All additional data are presented as **new Figures 1A-H** and **Supplementary figures 1D-J**.

3. Co-localization of Mfn2 and Tom20 is rare as shown by super-resolution microscopy. How can this be explained?

We agree that co-localization of Mfn2 and Tom20 in the original Figure 1B was inadequate. Images generated by super-resolution microscopy are computationally reconstructed and show dot patterns instead of line structures seen in confocal microscopy. In response, we have now replaced this data with new images to show their co-localization using super-resolution microscopy and parallel confocal microscopy data in **new Figures 1A-C** and **1F-H** in the revised manuscript. They clearly distinguish the mitochondrial Mfn2 (which is by far the bulk of Mfn2) and the newly discovered pool of plasma membrane junctional Mfn2.

4. Patterns of IF-staining of VE-cadherin and b-catenin (Fig-1F) and PLA analysis (2C) of same proteins do not correspond, thus raising concern about optimal conditions for PLA assay.

In response, we have now clarified the details of the proximity ligation assay (PLA) and conventional immunofluorescence and we realize that the original presentation of the data was confusing. Immunofluorescence shows the cellular location of VE-cadherin or β -catenin in the **new Figure 2A** (replaces the old Fig 1F).

However, the *in situ* proximity ligation assay (PLA) produces a spatial fluorescent signal (visualized as dots) within a 30-40 nm maximum distance and can thus determine close proximity interactions between protein partners^{1, 2}. If two proteins interact each other, the number of dots will be generated at interaction sites. Therefore, PLA assay in the previous Figure 2C (**new Figure 4E**) shows that the red dots under basal conditions are highly localized at the plasma membrane indicating interaction of VE-cadherin and β -catenin. The number of red dots at the plasma membrane decreases following TNF stimulation suggesting that inflammation induces dissociation of VE-cadherin and β -catenin.

5. Data of surface biotinylation assay are really confusing. B-catenin and Mfn2 are cytosolic proteins, and their biotinylation is a major concern. Presented decrease of biotinylated β -catenin and Mfn2 in TNF-treated cells does not provide any clue to jump to the conclusion that they are being released from assembled junctions.

The confusion primarily arose from the fact that we did not adequately explain the method of biotin labeling using a cell impermeable biotin. The biotin labels all cell surface proteins because we used a cell impermeable biotin, but all protein partners of biotinylated cell surface proteins are also pulled down even if these partner proteins bind cytosolic domains of the cell surface protein. Therefore, β -catenin (cytosolic) which is known to bind to VE-cadherin (cell surface) would be present in biotin pull-down samples even though β -catenin itself is not biotinylated. The presence of Mfn2 in these samples confirms that it interacts with a biotinylated cell surface protein, which we then show to be VE-cadherin. Furthermore, we could not detect Drp1, a mitochondrial fission protein in the same samples which shuttles between a cytosolic pool and mitochondria. However, the revised manuscript has several lines of evidence showing Mfn2- β -catenin-VE-cadherin interactions that there is no need to introduce the biotinylation data. We have therefore removed the cell surface biotinylation data in the revised manuscript.

6. *TNF α causes VE-cadherin internalization, which is inconsistent with surface biotinylation data in Fig-1D.*

We did not permeabilize the cells in the original Figure 1F which is why we could not infer VE-cadherin internalization. However, we now show that Mfn2 depletion disrupts adherens junctions (**new Figures 2A-B**). We analyzed area and fluorescence intensity of cell surface VE-cadherin in control and Mfn2-KD ECs using ImageJ. The data show that the area of VE-cadherin on the cell surface significantly decreased but there was no significant change in overall fluorescence intensity. We added the details of the experimental methods and present these additional analytic data in the revised manuscript (**new Figure 2B**).

We agree with the reviewer's comment about TNF α -induced VE-cadherin internalization that has been shown in previous reports. However, our cell surface biotinylation assay show that TNF α (10 ng/mL) did not induce VE-cadherin internalization in HLMVECs until 6 h (original Figure 1D). In addition, the protein levels of VE-cadherin did not decrease after TNF α treatment until 24 h by Western blotting assay (**new Supplementary figure 5G**). Although the total protein levels of VE-cadherin do not necessarily reflect VE-cadherin internalization as only a portion of internalized VE-cadherin is degraded and the rest undergoes recycling³, the timing of TNF treatment likely impacts the amount of internalized, recycled and degraded VE-cadherin. We now present the additional data for VE-cadherin protein levels after TNF α treatment in the revised manuscript (**new Supplementary figure 5G**).

7. *Blot reprobing with the antibody used for pulldown is essential as IP control in all co-IP expts.*

In response, we now show IP controls in all co-IP experiments in the revised manuscript.

8. *More definitive proof of Mfn2 nuclear localization is needed using 3D image reconstruction, given the reported close positioning of mitochondria and nucleus in certain conditions.*

In response, we have performed additional Mfn2 nuclear localization studies by showing 3D Z-stack images (**new Figures 7C-E**). We also have additional data to confirm nuclear Mfn2 by analyzing exogenously overexpressed GFP-Mfn2 and endogenous Mfn2 in basal conditions and following inflammatory activation. GFP-Mfn2 and endogenous Mfn2 are clearly translocated into nuclei by TNF α stimulation. We analyzed protein co-localization using ImageJ, 3D Imaris reconstruction, and ortho analysis to show their co-localization. The new data would clearly address these concerns and all additional data present at **new Figures 7C-E** in the revised manuscript.

9. *Fig-2A – western blot data not convincing and contradictory. No apparent difference of both, VE-cadherin and b-catenin association with Mfn2 between control and TNF-stimulated conditions. TNF causes dissociation of AJ complex and VE-cadherin internalization, which is not evident from presented data.*

We have restructured the manuscript to first show the baseline Mfn2 roles in the **revised Figure 1** and the Mfn2 roles during TNF α -stimulation in the **revised Figure 4** to clearly distinguish basal effects of Mfn2 depletion from those seen during inflammation. There is a clear interaction of Mfn2 and VE-cadherin or Mfn2 and β -catenin under resting conditions using immunoprecipitation in **new Figures 1I-J** in the revised manuscript. In addition, it was very clear that Mfn2 interacts with

β -catenin basally and their interaction was increased by TNF α stimulation in endogenous Mfn2 and exogenously Mfn2 overexpressed cells (**new Figures 4A-D**). However, interaction of Mfn2 and VE-cadherin did not be changed by TNF α stimulation in immunoprecipitation assay even though TNF α further decreased EC barrier integrity in Mfn2-KD ECs (**new Supplementary figures 4D-E**). We have moved the VE-cadherin data from the original Figure 2A to the **new Supplementary figure 4D-E**.

10. Fig-4B,D - western blot data are not convincing: protein changes are marginal, problematic to judge.

The original Figure 4B showed that Mfn2 translocate into nucleus at 6h after TNF α stimulation (upper band in doublet). Now, we indicate a correct band. We agree that TNF α induced β -catenin nuclear translocation appears low in the original Figure 4D even though it was statistically significant by quantification in the original Figure 4E. However, even low levels of nuclear β -catenin suffice for target gene activation⁴⁻⁶. We have now included NF- κ B as a well-known inflammatory transcription factor which is translocated into nuclei during inflammation (positive control) and Mfn1 (Mfn2 homolog that does not translocate) as a negative control. We present these additional data in the revised manuscript (**new Supplementary figure 6C**).

Minor points:

1. To better establish a barrier protective/anti-inflammatory role of Mfn2, it is essential to perform experiments with Mfn2 overexpression to test if such strategy rescues agonist-caused endothelial dysfunction.

In response, we performed rescue experiments for Mfn2 depletion by overexpressing GFP-Mfn2. We designed Mfn2 shRNA to target 3'-UTR regions of *Mfn2* gene instead of coding sequences (CDS), thus making GFP-Mfn2 impervious to Mfn2 shRNA. We found that the increased permeability in Mfn2-KD cells was significantly rescued by overexpressing GFP-Mfn2 (**new Figure 2D**). In addition, the disrupted barrier in Mfn2-KD ECs was also significantly restored in GFP-Mfn2 overexpressed Mfn2-KD ECs (**new Figures 2E and F**).

TNF α further disrupted the EC barrier in Mfn2-KD ECs (which already had partial barrier disruption). However, GFP-Mfn2 overexpression restored this additional TNF α induced EC barrier disruption in Mfn2-KD ECs (**Supplementary figure 4F**). These data strongly support that Mfn2 plays a critical role to maintain EC barrier integrity during homeostasis and inflammation. We present these additional data in the revised manuscript (**new Figures 2D-F and Supplementary figure 4F**).

2. The interpretation of Fig. 3C is unclear. How can it be confirmed that the two bands appearing in the absence of reducing agent are disulfide bond-linked Mfn2 and b-Catenin? Further experimental evidence is required.

The non-denatured proteins may have different 3D conformations and represent several bands of different sizes in non-reducing gels which may show smeared band ladders. Therefore, the complex of Mfn2 and β -catenin in non-reducing gels is shown mainly by double bands which may have undetectable different size bands in the original Figure 3C. We have additional data in which the bands of Mfn2 and β -catenin complex were increased by hydrogen peroxide (H₂O₂) treatment

in a time dependent manner, thus demonstrating that oxidation increases the Mfn2 and β -catenin interaction. We have replaced the original Figure 3C because the new data clearly shows the disulfide bond-linked Mfn2 and β -catenin. We present these additional data in the revised manuscript (**new Supplementary figures 5A-B**).

3. *Fig. 1B needs to be mentioned in the text. Similarly, there are some phospho-beta catenin blots in supplementary figs, which also need to be mentioned.*

We have replaced Fig.1B with higher quality images in the **revised Figure 1** and we now show phospho- β -catenin data in **new Supplementary figures 5E-F**.

4. *Was there any specific reason to use doxycycline-inducible lentiviral shRNA for Mfn2 knockdown? Why regular siRNA transfection cannot be used?*

Some of the main advantages are the lentiviral vector efficiency of shRNA and the fact that the doxycycline-inducible lentiviral shRNA system produces cells with same genetic background all are transfected with the same construct and the only difference is +/- doxycycline. However, we also have additional data using Mfn2 siRNA on EC junctional integrity and we have now also compared Mfn2 siRNA and Mfn1 siRNA in parallel. We present these additional data in the revised manuscript (**new Supplementary figures 2A-D**).

5. *Fig. 3F, mice were given 8mg/kg of LPS i.p. This is a sub-optimal model of lung injury.*

We agree with reviewer that this is a sublethal dose (8 mg/kg i.p.) which induces mild lung injury with 10% lethality (1 mouse among 10 mice within 48h). We wanted to emulate non-lethal inflammation and assess if this was sufficient to induce the sulfenylation changes in the live but inflamed mice.

Reviewer #2 (Remarks to the Author):

“The authors report a set of observations related to the role of Mfn2 in adherens junctions and related inflammatory response. This interesting role, previously unreported, is based on several different measurements. I have primarily focused on the microscopy here, since it is my main expertise.”

1. *It is important to note that Alexa 647 contains an impurity that fluoresces in the near-red (Stone MB, Veatch SL. Far-red organic fluorophores contain a fluorescent impurity. Chemphyschem. 2014;15(11):2240–2246. doi:10.1002/cphc.201402002). So, the authors’ use of Alexa647 together with Alexa595 to look for co-localization (Fig. 1A) is unfortunate, since it may be influenced by this impurity. This is underlined here, where the “colocalization” signal in the 595 channel is very weak (low SNR). To demonstrate co-localization, the authors should use additional pairings of labels, such as using Mfn2-GFP and a near-red dye for mitochondria together with VE-cadherin Alexa647, and the pair of Mfn2-GFP with VE-cadherin Alexa647 alone.*

We agree that Alexa647 contains an impurity in the near-red wavelengths. Mfn2 and VE-cadherin were detected with Alexa488 and Alexa647, respectively. Thus, the “co-localization” signal in the A488 channel for Mfn2 within the A647 channel for VE-cadherin did not contain any impurity,

suggesting co-localization of Mfn2 and VE-cadherin (original Figure 1A). Tom20 was detected with Alexa595 which may have an impurity signal with VE-cadherin detected with Alexa647.

In response to the reviewer's suggestions, we performed new studies and examined different combinations of secondary antibodies as follows; **1)** Mfn2 (A568 or A546): Tom20 (A488): VE-cadherin (A647), **2)** Mfn2 (A488): Tom20 (A405): VE-cadherin (A647), **3)** Mfn1 (A488): Tom20 (A568): VE-cadherin (A647), **4)** Mfn1 (A546): Tom20 (A488): VE-cadherin (A647), **5)** Tom20 (A568): VE-cadherin (A488), **6)** Tom20 (A568): VE-cadherin (A647).

We could avoid potential artifacts from "impurities" with different wavelengths by using secondary antibodies with other wave lengths. We found that the Tom20 signal (using A405) was much weaker than in the images shown (original Figure 1A). Therefore, we did not use secondary A405 antibody. Finally, we have obtained some of this data and additional images using confocal and super-resolution microscopy to further increase the robustness of the findings. We now show a clear mitochondrial structure with Mfn2, Tom20, and Mfn1 staining. The new data in the revised manuscript clearly address these concerns. All additional data are presented in **new Figures 1A-H** and **Supplementary figures 1D-J**.

2. The data referenced to show that Mfn2 does not necessarily colocalize with mitochondria (SFig 2A) are not convincing. Under inflammatory conditions, there is still a colocalization, which becomes more obvious when increasing the gain. Also a quantification for colocalization is missing.

We have clarified that the vast majority of Mfn2 co-localizes with mitochondria (specifically, we used the mitochondrial membrane protein Tom20) as this is the primary Mfn2 pool and its canonical function. Our novel findings focus on the new non-canonical role of Mfn2 which does not co-localize with mitochondria and is instead found at the junctions. Even during inflammation, this pool of non-canonical Mfn2 remains small but translocates to the nucleus. We have moved the original Supplementary Figure 2A to the **new Figure 1D** and added a quantification for colocalization between GFP-Mfn2 and Tom20 using ImageJ at **new Figure 1E**.

3. The authors cite the expected resolution achievable by SIM of 130 nm, but they do not seem to measure it for their images. A quick check shows that at least for some of the data included the pixel-based resolution is quite low, ~300 nm. However, the claimed resolution may not be necessary for the conclusions drawn.

The super-resolution microscopy (SIM) data was acquired by a GElifeSciences OMX system with the proprietary AcquireSR program (Version: 4.4.9800-1) which provides high spatial resolution (lateral resolution ~120 nm). Reconstruction and alignment were carried out by the Deltavision softWorRx (version: 7.0.0) software. The final reconstructed file (Tif) for each location and condition is about 350GB containing about 28-30 Z-planes. For this reason, we created a sub-stack and each of the submitted sub-stack contained 3 Z-planes. The figures (JPG) presented in original Figure 1C (new Supplementary figure 1D) are based on these image stacks. In the revised manuscript, we have added new projection images from 3D-SIM data containing all Z planes in **new Figures 1A** and **1F**.

4. Overall, the staining of mitochondria is of low SNR, to the extent that it is sometimes difficult to recognize individual mitochondria (Figure 1A).

The images generated by super-resolution microscopy (SIM) were created by computational reconstruction (original Figure 1A). In response, we have replaced original Figure 1A and added new images as well as quantification data to show their co-localization using parallel super-resolution microscopy and confocal microscopy imaging in **new Figures 1A-H** and **Supplementary figures 1F-J** in the revised manuscript. These data now clearly show mitochondrial structure as well as non-mitochondrial Mfn2.

5. The image thresholds appear to have been changed, resulting in intensity saturation (enlargements, Fig. 1B).

Images from super resolution microscopy were analyzed for 3D imaging by using Imaris software which may affect the appearance of image thresholds. In response, we have replaced this data with the newer confocal and super-resolution microscopy data in the **revised Figure 1**.

6. Scale bars seem to be wrong in Panels 1B, 1C, 2C, 4F (not consistent between different magnifications), missing in 2E.

We agree with the reviewer and have corrected scale bars for all images. Some of the differences were due to using distinct software such as the proprietary Deltavision softWorRx (version: 7.0.0), Imaris, Zen blue, or ImageJ software packages.

7. Figure 1A and 1C show identical panels

We have revised these panels in the revision. We removed original Figure 1A and basal part of original Figure 1C was moved to new Supplementary figure 1D.

8. The disruption of adherens junctions upon Mfn2 knockdown is striking (Fig. 1F), but should be quantified.

In response, we have now analyzed area and fluorescence intensity of cell surface VE-cadherin and β -catenin in control and Mfn2-KD ECs (original Figure 1F) using ImageJ. The data show that the area of AJs proteins on cell surface significantly decreased but not fluorescence intensity. We present these additional data in the revised manuscript (**new Figure 2B**).

9. Differences upon TNF-alpha treatment appear convincing (2E). However, the quantification of differences does not appear to have been done on a sufficient sample size: Fig 2D and F each show only 6 data points each, not enough to know how the data is distributed.

We used interaction of VE-cadherin and β -catenin as a positive control for proximity ligation assay because they are well-known interacting partners in ECs⁷. We found that the interaction of VE-cadherin and β -catenin in one field generated average 450 dots (interactions) in basal condition. We analyzed 6 different fields from different cells and the results were significantly different between basal and TNF α treated cells (original Figure 2D) to serve as a positive control. The interaction of Mfn2 and β -catenin was analyzed as numbers of dots (interactions) per cells and the results show that TNF α treated cells generated average 25 dots which are five-fold increased

than control cells. We analyzed 104 cells and 212 cells in basal condition and TNF α treated cells, respectively. Based on the number of analyzed cells, we considered the sample size sufficient.

10. For some datasets, I could not find a description of how the analysis was performed (for example Fig. 2D, F “number of interaction”).

In response, we provide more details about the proximity ligation assay: *The red/green dots show an interaction between VE-cadherin and β -catenin, or Mfn2 and β -catenin. Quantification of the number of interactions (dots) per field.*

11. For statistics, the authors described that they have used student’s t-test for comparisons. However, if the data are not normally distributed, a non-parametric test has to be used. For example, data in Figures 2D, 2F, 3B, 3C, 4G should be checked for whether they are normally distributed.

In response, we tested the frequency distribution according to the reviewer’s comments and present them as **reviewer only Figure 2** showing that the data points are normally distributed.

We have included a more detailed explanation about our statistical analysis in the revised manuscript as following; **“Statistical analysis.** The bands by Western blotting were analyzed for optical density using ImageJ (NIH) software. Colocalization of fluorescence images by confocal microscopy were analyzed by evaluating Manders’ coefficient or Spearman’s correlation using ImageJ. Quantification of replicate experiments is presented as the mean \pm SEM from at least 3 independent experiments. The student t-test, one-way and two-way ANOVA with Bonferroni post-tests were used to determine statistical significance, with a *p* value threshold of less than 0.05. Significance levels are indicated in the figures as **p* < 0.05, ***p* < 0.01, ****p* < 0.001, and *****p* < 0.001. No statistical methods were used to predetermine sample sizes, but our distribution was assumed to be normal and variances were assumed to be equal across groups, but this was not formally tested. All analyses were conducted using Prism, GraphPad Software (La Jolla, CA).”

Reviewer #3 (Remarks to the Author):

“This is a very interesting novel concept and the title is a fair assessment of their findings. However, some main conclusions are not fully supported by the data presented. First, the authors imply a non-

mitochondrial role of Mfn2, instead depending on a cytoplasmic and nuclear translocation of soluble Mfn2. In fact, Mfn2 is a transmembrane protein and thus it is difficult to envision how the full length protein could be soluble. Therefore, they need to either definitely exclude a role of mitochondria or revise their model. Equally, a direct effect of nuclear (soluble?) Mfn2 on beta-catenin transcription should be either fully addressed or the model should be revised.”

In this manuscript, the authors investigate the role of mitofusin2 (localised to the outer membrane of mitochondria) in adherent junction complexes (localised at the plasma membrane), focussing on their importance in endothelial barrier and transcriptional responses upon inflammatory stimulation. They convincingly demonstrate (in cells and in mice lungs) a totally novel role of Mfn2 (or at least of mitochondrial morphology) in restricting endothelial permeability. Consequently, Mfn2 prevents beta-catenin-dependent induction of pro-inflammatory genes, proposed to occur via oxidative-dependent modification of beta-catenin. They observe that Mfn2 localises proximal to the plasma membrane, via association with VE-cadherin under basal conditions. Upon TNFalpha stimulation, beta-catenin dissociates from VE-cadherin and further associates with Mfn2. This is a very interesting novel concept and the title is a fair assessment of their findings. However, some main conclusions are not fully supported by the data presented. First, the authors imply a non-mitochondrial role of Mfn2, instead depending on a cytoplasmic and nuclear translocation of soluble Mfn2. In fact, Mfn2 is a transmembrane protein and thus it is difficult to envision how the full-length protein could be soluble. Therefore, they need to either definitely exclude a role of mitochondria or revise their model. Equally, a direct effect of nuclear (soluble?) Mfn2 on beta-catenin transcription should be either fully addressed or the model should be revised.

Main points:

Mfn2 depletion disrupts AJ junctions. But what is really the role of Mfn2? The authors propose that Mfn2 interaction with VE-cadherin and beta-catenin in presence/absence of TNF alpha require its re-location outside mitochondria. However, this must be convincingly demonstrated (see technical comments below).

Or, instead, could beta-catenin signalling depend on the presence of mitochondria at the plasma membrane, where Mfn2 would act as a tether between both “organelles”? Alternatively, could it instead depend on mitochondrial morphology, rather than on Mfn2 itself? Does it depend on Mfn1 or Opa1? Can it be suppressed by restoring tubulation, e.g. upon depletion of both fusion and fission components?

We also initially thought that mitochondria might translocate to the plasma membrane, resulting in the localization of Mfn2 at the plasma membrane. However, our results indicate that only Mfn2 co-localizes with VE-cadherin which is a well-known EC adherens junction plasma membrane protein but not other mitochondrial proteins (such as the routinely used mitochondrial membrane protein Tom20 or the Mfn2 homolog Mfn1). We now have robust additional data Mfn2 localization at the plasma membrane using super-resolution microscopy and confocal microscopy in **new Figures 1A-H** and **Supplementary figures 1D-J**. We have also added rescue effects of Mfn2 to stabilize EC barrier integrity and function by overexpressing shRNA resistant GFP-Mfn2 in Mfn2-KD ECs at **new Figures 2D-F** and **Supplementary figure 4F**. These supporting results suggest that Mfn2 itself can localize at plasma membrane independent of mitochondria to stabilize AJs junctional integrity under resting condition. We believe that showing the Mfn2 specificity of our findings (and the absence of Mfn1) strongly bolsters our original findings.

However, we agree that Mfn2 may also tether mitochondria and the plasma membrane similar to the reported role for mitochondria and ER tethering at mitochondria associated ER membrane (MAM)⁸. We have added this discussion point and reference.

The nuclear localization of Mfn2 (Fig. 4, see detailed technical comments below) needs to be convincingly demonstrated. Instead, does Mfn2 interact with beta-catenin and with nuclei acid binding proteins in the cytoplasm? Indeed, externalization to the cytoplasm of both nuclear and mitochondrial DNA has been recently shown. An experiment similar to the one presented in Sup1 instead using nuclear/PM/mitochondrial and soluble fractions should allow clarifying this point.

We found that a subset of Mfn2 translocates into the nucleus following TNF stimulation using immunofluorescence and subcellular fractionation assay. The nuclear Mfn2 inhibits β -catenin transcriptional activity in the nucleus during inflammation using a luciferase activity assay. We provide additional new data for nuclear Mfn2 by showing co-localization analysis, 3D images, ortho analysis in 3D-Z sectioned images using GFP-Mfn2 and endogenous Mfn2 in the **new Figures 7C-E**.

The authors convincingly demonstrate formation of disulphide bonds in beta-catenin upon oxidative stress, which co-precipitate with Mfn2. However, they state without demonstrating that it is an S-S link between Mfn2 and beta-catenin. Moreover, whether beta-catenin S-S bond is relevant or depends on Mfn2 is not analysed.

Our immunoprecipitation data used a Mfn2 specific antibody show that Mfn2 interacts with β -catenin dependent on ROS under reducing (with β -mercaptoethanol) and non-reducing (without β -mercaptoethanol) gels. Under nonreducing condition, the shifted bands (between 150 and 250 KDa) are formed by S-S link between Mfn2 (78 KDa) and β -catenin (95 KDa). However, if either β -catenin or Mfn2 itself forms S-S linkage, the bands may show above 250 KDa size because (Mfn2 (78 KDa)+ β -catenin-S-S- β -catenin (95+95=190 KDa)= 268 KDa) and 251 KDa (β -catenin (95 KDa)+ Mfn2-S-S-Mfn2 (95+95=156 KDa)) KDa, respectively. Therefore, our result suggest that Mfn2 interacts with β -catenin via disulfide formation under inflammation.

The non-denatured proteins may have different 3D conformations and represent several bands of different sizes in non-reducing gels which may show smeared band ladders. Therefore, the complex of Mfn2 and β -catenin in non-reducing gels is shown mainly by double bands which may have undetectable different size bands at original Figure 3C. In response to the reviewer's suggestion, we now show additional data in which the bands of the Mfn2/ β -catenin complex are increased by hydrogen peroxide (H_2O_2) treatment in time dependent manner. This data now replaces the prior Figure 3C because this new data more clearly shows that the disulfide bond-linked Mfn2/ β -catenin depend on the redox status. We present these additional data in the revised manuscript (**new Supplementary figures 5A-B**).

Further technical / clarity concerns:

They could broaden the references mentioned in the introduction and include more review papers for a non-specialist audience. The reason for specifically studying Mfn2 should be added. A review reference for the pro- or anti-fusion roles of Mfn2 in mitophagy should be included. Reference 19 is controversially discussed and this should be mentioned.

We have revised introduction according to reviewer's comments. We address more general background, specific roles about Mfn2 in the revised manuscript.

Fig. Sup1: The raw data with the identity and quantification of all Mfn2 interactors depicted in S1B-C found should be included. This is essential to assess the relevance of their further studies. Importantly, Mfn2 is mostly facing the cytoplasm, thus interaction with non-mitochondrial proteins does not allow to exclude that it does so while still located to mitochondria.

In response, we have included lists of all Mfn2 interacting partners in **new Supplementary Figure 1C**. We did not exclude Mfn2 binding partners which are located to mitochondria. We were interested in the surprising non-canonical roles of Mfn2, hence the focus on non-mitochondrial proteins as binding partners of Mfn2.

Fig. Sup 2A: Absence of co-staining between Tom20 and Mfn2 does not allow excluding Mfn2 from mitochondria. This is a critical point that needs to be convincingly demonstrated. Co-immuno-staining but also biochemical fractionation with other mitochondrial markers (OM, matrix, IM and IMS-located) must be performed. How specific is the Mfn2 antibody or siRNA? Does it also affect Mfn1?

In the original Supplementary figure 2A, ECs overexpressing GFP-Mfn2 (green fluorescent protein conjugated Mfn2) were stained with a Tom20 antibody. The images showed that GFP-Mfn2 mainly co-localized with Tom20 (the expected mitochondrial pool of Mfn2 which is the vast majority Mfn2), but some Mfn2 complex did not co-localize and indicated surprising non-mitochondrial Mfn2. We did not use any antibody for Mfn2 because this experiment relied on lentiviral GFP-Mfn2.

In addition, we now have additional data of co-staining between Mfn2 and Tom20 using super-resolution microscopy in the **new Figure 1A**. We used two different antibodies for each Mfn2 and Tom20. We already confirmed that our shRNA and siRNA for Mfn2 specifically targets Mfn2 without affecting to Mfn1. We have also included additional data about the specificity of Mfn2 sh/siRNA at **new Supplementary figures 2A-B** in the revised manuscript.

Fig. 1: Include co-staining between Mfn1, Mfn2 and VE-cadherin, and also between Mfn2, Tom20 VE-cadherin upon TNF alpha. Figure 1B is not mentioned in the text. In 1D include a negative control, e.g. Tom20, Mfn1 or other interacting proteins from the pool-down shown in sup1. In 1F, how does Mfn2-KD change the localization of VE-cadherin and beta-catenin upon TNF alpha, and how does this affect the localization of mitochondria? This is important to interpret the biological significance of the results shown in Fig. 2 C-D and E-F. Also, in 1F, include Mfn2 rescue experiments.

In response, we have performed new experiments with data of co-staining between Mfn2, Tom20, and VE-cadherin under resting condition at **new Figures 1A-C**. We also have additional data of co-staining between Mfn1, Tom20, and VE-cadherin under resting condition in **new Figures 1F-H** and **Supplementary figures 1F-J**. In addition, we added biochemical analysis to show their interaction in **new Figures 1I-J**. We reconstructed figures to present the findings more clearly. Thus, we moved part of original Figure 1C to Supplementary figures 1D-E and removed original Figure 1B.

We agree with the reviewer that the original Figure 1D using cell surface biotinylation would benefit from controls. We have a negative control, Drp1 which is cytosolic but not at cell surface (**reviewer only Figure3**). It clearly shows Mfn2 interacts with cell surface proteins such as VE-cadherin but not Drp1. However, as indicated by other reviewers, it may be confusing for readers to understand why we show surface biotinylation data even though Mfn2 interacts with the cytosolic domain of VE-cadherin. We have therefore removed the biotinylation data from the manuscript but show it for the benefit of the reviewers in this reviewer response. Instead of the biotinylation data, we now substantially expanded the imaging assays with 3D Z-stacks to convincingly show plasma membrane and nuclear localization of Mfn2.

We agree that this new role of Mfn2 at the plasma membrane raises many potentially exciting questions about its roles. We therefore examined whether Mfn2 mediates interaction between F-actin and VE-cadherin to stabilize AJs junctional integrity because F-Actin is a known stabilizer of AJs^{9, 10} and our proteomics data also identified an actin-binding protein as a Mfn2 partner. We found that Mfn2-KD ECs showed marked decreases in F-actin-VE-cadherin association in the **new Supplementary figures 2F-H**. This does not rule out additional functional roles of non-canonical Mfn2 at the plasma membrane.

Fig. Sup 2B: include Mfn1 control and Mfn2 rescue experiments. Lines 150-153: the text does not accurately describe the experiments performed.

In response, we have included Mfn1 control at original Supplementary Figure 2B (**new Supplementary Figure 2A** in revised manuscript). We also added Mfn2 rescue experiments for EC barrier integrity and cell permeability at **new Figures 2D-F**. We overexpressed Mfn2 shRNA resistant GFP-Mfn2 into Mfn2-KD ECs and examined whether GFP-Mfn2 can restore the disrupted EC barrier and increase cell permeability in Mfn2-KD ECs. Rescue experiments strongly support a role of Mfn2 as a AJs stabilizer in ECs. We have these additional data at **new Figure 2D-F** in the revised manuscript and the text has been revised.

Sup3B: How specific to Mfn2 is the GTPase activity observed? Include the value with IgG control.

It has been appreciated that we can measure GTPase activity after purification by immunoprecipitation^{7, 11}. Therefore, in response to the reviewer comments, we immunoprecipitated Mfn2 and used a negative control IgG as presented in the **new Supplementary figure 5D**. The immunoprecipitated Mfn2 was used for a GTPase assay using the colorimetric GTPase assay kit (NUVUS, Innova Bioscience, 602-0120) according to manufacturer guideline. Mfn2 GTPase activity in samples was normalized by subtracting the value generated by IgG sample and the relative values as percent (%) was presented compared to the control (TNF α 0 h) at **new Supplementary figure 5C**.

Fig. 2E: draw also here the boundaries, similar to 2C.

We have the boundaries into original Figure 2E (**new Figure 4G**).

Lines 175-176: it is not necessarily the same Mfn2 population that interacts with VE-cadherin and beta-catenin in presence and absence of TNS alpha. I recommend to tone down.

We agree with reviewer's comment and have modified it as follows: "Based on multiple lines of inquiry, we concluded that Mfn2 disassociates from VE-cadherin and there is a concomitant increase of Mfn2 interaction with β -catenin in the cytosol following inflammatory stimulation."

Fig. 3: It is very clear that Mfn2 - beta-catenin interaction depends on ROS. However, there is no proof that Mfn2 is forming a S-S link with beta-catenin. Thus, in 3C, perform IB also with Mfn2. Further, analyse beta-catenin S-S formation in Mfn2-KD. Moreover, to prove their claim that interaction is mediated by S-S formation (lines 177-178), they would need to identify the responsible cysteines and observe abolished inflammatory responses in the respective mutants. In the same line, the conclusion presented (lines 199-202) should be proven or toned down.

The reviewer mentioned this also as a major comment and we have addressed it above in the **new Supplementary figures 5A-B**.

We have also included additional data demonstrating that the interaction between Mfn2 and β -catenin was decreased in Mfn2-KD ECs in **new Supplementary figure 1K**.

For Lines 199-202, we have the revised sentence as following: "These data suggest that during inflammation, ROS mediate the interaction of Mfn2 and β -catenin via disulfide bond formation."

Fig. 4 and Sup 4B: As stated above, the claimed nuclear translocation of Mfn2 must be convincingly demonstrated or eliminated from the manuscript. After all, their experiments do not address what Mfn2 would do in the nucleus. In 4A,F: Statistical analyses for Mfn2 must be included, other mitochondrial proteins (e.g. Mfn1, Tom20, mitotracker) must be tested. In fact, in 1F, "nuclear" localization of Mfn2 is also visible under basal conditions. In 4B: Simultaneous fractionation of mitochondrial, plasma membrane, nuclear and soluble fractions must be included, analysed by respective markers (including Mfn1, Tom20 and other mitochondrial proteins and also including beta-catenin and other cytosolic-nuclear proteins known to translocate upon TNF alpha. How do they explain that "nuclear" increase is much stronger for Mfn2 than for beta-catenin (Fig. 4B-E)? In fact, the nuclear translocation of beta-catenin is also not very

strongly observed by cellular staining, in Sup 4B. It rather appears that it is perinuclearly accumulating in the cytosol. Thus, the interaction of both Mfn2 and Sup 4B with nucleic acids could occur outside the nucleus.

As we mentioned above, it is possible that Mfn2 interacts with β -catenin and with nucleic acid binding proteins in the cytoplasm. However, we found that a subset of Mfn2 translocates into the nucleus by TNF α stimulation using immunofluorescence and subcellular fractionation assay. Mfn2 inhibits β -catenin transcriptional activity in the nucleus during inflammation as shown by a β -catenin luciferase activity assay (**new Figures 6C-D**). We provide additional evidence for TNF α induced Mfn2 nuclear translocation by co-localization analysis, 3D images, ortho analysis in 3D-Z stack images with GFP-Mfn2 and endogenous Mfn2 at **new Figures 7C-E**. We also examined whether TNF α induced nuclear translocation of the Mfn2 homolog Mfn1 and the well-known inflammatory transcription factor, NF-kB which is translocated during inflammation. We found that NF-kB clearly translocated into the nucleus following TNF α stimulation but not Mfn1 using a subcellular fractionation assay. Thus, we have added NF-kB data as a positive control and Mfn1 as a negative control for subcellular fraction assay. We present these additional data in the revised manuscript (**new Supplementary figure 6C**).

It has also been appreciated that low levels of nuclear β -catenin suffice for target gene activation⁴⁻⁶. Along the same line, our β -catenin luciferase assay in the new Figures 6C-D indicates that nuclear β -catenin activity was inhibited by Mfn2 under TNF α stimulation. We believe that β -catenin activity may be a more robust indicator of how Mfn2 acts. Comparisons of Mfn2 levels and β -catenin by Western blotting depend highly on the antibody affinity and cannot necessarily be directly compared after cell fractionation. We have moved the original Figures 4D-E to the new Supplementary figures 6C-D in revised manuscript and focus on the multiple advanced imaging analyses and β -catenin activity in the main manuscript.

Fig. 5A: For VCAM-1 there is no Mfn2 dependence and it is not mentioned in the text. It could be eliminated.

We agree with reviewer's comment and have removed it in the revised manuscript.

There are several typos throughout the text and lost numbers within the sup figures. Please correct.

We appreciate reviewer's detail comments and corrected typos in text and revised all numbers within the Supplementary figures.

Reference

1. Rico, A.J. *et al.* Neurochemical evidence supporting dopamine D1-D2 receptor heteromers in the striatum of the long-tailed macaque: changes following dopaminergic manipulation. *Brain Struct Funct* **222**, 1767-1784 (2017).
2. Dasgupta, S. *et al.* Metabolic enzyme PFKFB4 activates transcriptional coactivator SRC-3 to drive breast cancer. *Nature* **556**, 249-254 (2018).
3. Juettner, V.V. *et al.* VE-PTP stabilizes VE-cadherin junctions and the endothelial barrier via a phosphatase-independent mechanism. *J Cell Biol* **218**, 1725-1742 (2019).
4. Li, V.S. *et al.* Wnt signaling through inhibition of beta-catenin degradation in an intact Axin1 complex. *Cell* **149**, 1245-1256 (2012).

5. Goldstein, B., Takeshita, H., Mizumoto, K. & Sawa, H. Wnt signals can function as positional cues in establishing cell polarity. *Dev Cell* **10**, 391-396 (2006).
6. Nusse, R. & Clevers, H. Wnt/beta-Catenin Signaling, Disease, and Emerging Therapeutic Modalities. *Cell* **169**, 985-999 (2017).
7. Dejana, E. & Orsenigo, F. Endothelial adherens junctions at a glance. *J Cell Sci* **126**, 2545-2549 (2013).
8. Filadi, R., Pendin, D. & Pizzo, P. Mitofusin 2: from functions to disease. *Cell Death Dis* **9**, 330 (2018).
9. Meng, W. & Takeichi, M. Adherens junction: molecular architecture and regulation. *Cold Spring Harb Perspect Biol* **1**, a002899 (2009).
10. Hayer, A. *et al.* Engulfed cadherin fingers are polarized junctional structures between collectively migrating endothelial cells. *Nat Cell Biol* **18**, 1311-1323 (2016).
11. Kim, Y.M. *et al.* Redox Regulation of Mitochondrial Fission Protein Drp1 by Protein Disulfide Isomerase Limits Endothelial Senescence. *Cell Rep* **23**, 3565-3578 (2018).

Reviewers' comments:

Reviewer #1 (Remarks to the Author):

The authors performed additional experiments and made essential clarifications in the manuscripts. They also improved the quality of presented data and provided essential controls. Overall, they addressed major concerns by this reviewer

Reviewer #2 (Remarks to the Author):

Overall, I found the quality of the images (signal to noise ratio) to be improved, duplications removed and scale bars corrected. This takes care of my points 1, 2, 4, 5, 6, 7. Additional information/analysis has been added (points 8, 9, 10).

This leaves only points 3 and 11. For 3, the authors insist on the microscopy specifications, but that does not mean that the data achieves this resolution. All of the images and quantifications are zoomed out to a scale where superresolution is not needed. Thus, I still object to the statement on line 113 of the 120 nm lateral resolution which is not demonstrated in any images or line profiles throughout the manuscript.

For point 11, the histograms included in the rebuttal letter do not show that the distributions are normal. IT IS DIFFICULT TO TELL WHAT THE DISTRIBUTION IS FOR SMALL SAMPLE SIZES. There are different ways to do this, here is an excerpt from Wikipedia: "An informal approach to testing normality is to compare a histogram of the sample data to a normal probability curve. The empirical distribution of the data (the histogram) should be bell-shaped and resemble the normal distribution. This might be difficult to see if the sample is small. In this case one might proceed by regressing the data against the quantiles of a normal distribution with the same mean and variance as the sample. Lack of fit to the regression line suggests a departure from normality"

Reviewer #3 (Remarks to the Author):

Review of NCOMMS-19-32607A-Z

As before, even more convincingly shown now, a fraction of Mfn2 does not co-localize with Tom20 but instead co-localizes with adherens junctions. Moreover, Mfn2 interacts physically with beta-caterin, which increases upon oxidation. Importantly, Mfn2 is required for EC barrier integrity and protects from inflammation. These physiological consequences have also been improved and are supported by the data. As stated before, this is very interesting and would deserve publication. However, beyond these findings, the mechanistic conclusions taken throughout the manuscript, also placed in the final model, are not supported by the data and must be eliminated. In this respect, as detailed below, the revision is very disappointing. From my main conceptual concerns none has been correctly addressed and the responses to the technical points were only poorly answered and in some cases are even misleading.

1- Mfn2 depletion disrupts AJ junctions

As stated before, this could depend on the mitochondrial morphology defect specifically

caused by Mfn2 depletion. The request to analyse EC barrier integrity upon depletion of Mfn1, Opa1, Drp1 and both fusion and fission components was not addressed.

2- A fraction of Mfn2 does not co-localize with Tom20.

As stated before, absence of Mfn2 co-staining with Tom20 does not exclude Mfn2 from mitochondria and does not exclude the presence of mitochondria in close proximity to adherens junctions. The fact that Mfn1 does not locate to AJ junctions is not a valid argument to exclude Mfn2 from the mitochondrial membrane. The request to analyse other mitochondrial markers (OM, IN, IMS, matrix) and to perform biochemical fractionation of mitochondria was not addressed.

3- The nuclear localization of full length Mfn2 (visible from their western blots) is not convincingly demonstrated. As state before, Mfn2 is an integral membrane protein. It will not "simply" translocate to the nucleus (upon disruption of the nuclear barrier, Mfn2 could e.g.be close to DNA in the cytoplasm, be inserted in vesicles present in the nucleus,...). However, the biochemical fractionation into soluble vs membrane proteins was not addressed.

4- A disulphide bond between Mfn2 and beta-caterin is not demonstrated. Showing that cysteines from both Mfn2 and beta-caterin are available for sulfenylation and that their physical interaction increases upon H₂O₂ does not prove it. Beta-caterin size clearly shifts up upon oxidation, in non-reducing gels, but whether this depends on Mfn2 was not analysed. Importantly, and in contrast to beta-caterin, there is no convincing shift observed for Mfn2 in Sup fig 5. It rather suggests that oxidation leads to beta-caterin S-S formation(s) (this could easily be tested), which increases its binding to Mfn2, perhaps by exposing a bigger/better interaction interface. The request to identify the responsible cysteines was not addressed (although mass-spec analysis of their DCP-bio experiments should allow it).

Further, from the previously raised technical and clarity concerns, just to mention a few:

1- I found no new review Mfn2 references, like for example PMID 32304672, 31156466, 31252211. The findings ascribed in references 13,14,15 are not relevant for this manuscript. Mfn2 clearly regulates the proximity between mitochondrial and the ER. However, regarding the localization of Mfn2 at the ER, if mentioned, it must be stated that it is controversial. Also, while being both extremely relevant, refs 18,19 are not the most appropriate for the localization statement.

2- The raw data was not provided. Just showing a selected name list does not replace for the requested identity and quantification of the proteins found.

3- Regarding GTPase activity, it was not convincing before. What the authors call new sup fig 5C and D was already there before as sup fig. 3 B and C. Thus, showing exactly the same panels, does not help and is inappropriate.

Finally, some of the new data presented lacks controls and is overstated.

Point-by-point responses:

Reviewer 1 indicated that our last revision had addressed all the previously raised points. Reviewers 2 and 3 listed some additional points that asked us to address. We have now added additional data and analyses, as well as made changes in the text in response to the comments of reviewers 2 and 3. We quote the revised manuscript text and new results in the responses below. We also provide a list of new references that have been added to the manuscript. We have also adjusted the cited reference numbers in this response document so that it is easier for the reviewers to review the new references. For the manuscript to remain manageable for the reader, we have included some of the new data that specifically addresses reviewer comments in this response document as Reviewer Figures (R1, R2, etc).

Reviewer #2 (Remarks to the Author):

This leaves only points 3 and 11. For 3, the authors insist on the microscopy specifications, but that does not mean that the data achieves this resolution. All of the images and quantifications are zoomed out to a scale where super resolution is not needed. Thus, I still object to the statement on line 113 of the 120 nm lateral resolution which is not demonstrated in any images or line profiles throughout the manuscript.

The reviewer is correct, the images we show do not need the 120 nm resolution of the super resolution microscope. We have therefore removed the statement regarding the 120 nm resolution from the revised manuscript.

For point 11, the histograms included in the rebuttal letter do not show that the distributions are normal. IT IS DIFFICULT TO TELL WHAT THE DISTRIBUTION IS FOR SMALL SAMPLE SIZES. There are different ways to do this, here is an excerpt from Wikipedia: "An informal approach to testing normality is to compare a histogram of the sample data to a normal probability curve. The empirical distribution of the data (the histogram) should be bell-shaped and resemble the normal distribution. This might be difficult to see if the sample is small. In this case one might proceed by regressing the data against the quantiles of a normal distribution with the same mean and variance as the sample. Lack of fit to the regression line suggests a departure from normality"

We agree with the reviewer that a formal statistical test to assess for normal distribution should be used prior to performing the statistical analysis and thus choosing the appropriate test, depending on whether or not the data is normally distributed. We use the Shapiro Wilk test which is suited for assessing normal distribution in smaller

Figure	Shapiro-Wilk test	W	P value	Passed normality test (alpha=0.05)?	P value summary
Fig4F		0.7995	0.9545	Yes	ns
Fig4H		0.9361	0.9632	Yes	ns
Fig5B		0.8929	0.7963	Yes	ns
SupFig5B		0.8929	0.9889	Yes	ns
SupFig6F		0.5984	0.9053	No	**

Reviewer Figure R1. Test for normal distribution of data using the Shapiro-Wilk test.

samples¹. As shown in **Reviewer Figure R1**, the quantitative data was normally distributed except for the data in Supplementary Fig S6F.

We have revised the methods section as follows “Data distribution was analyzed with the Shapiro-Wilk test¹ to assess for normal distribution. For normally distributed data, the Student t-test, one-way and two-way ANOVA with Bonferroni post-tests were used for between group comparisons to determine statistical significance, with a p value threshold of less than 0.05. When data were not normally distributed, we used the non-parametric Mann-Whitney test for between group comparisons. Significance levels are indicated in the figures as *p < 0.05, **p < 0.01, ***p < 0.001, and ****p < 0.0001.”

Reviewer #3 (Remarks to the Author):

Review of NCOMMS-19-32607A-Z

As before, even more convincingly shown now, a fraction of Mfn2 does not co-localize with Tom20 but instead co-localizes with adherens junctions. Moreover, Mfn2 interacts physically with beta-catenin, which increases upon oxidation. Importantly, Mfn2 is required for EC barrier integrity and protects from inflammation. These physiological consequences have also been improved and are supported by the data. As stated before, this is very interesting and would deserve publication. However, beyond these findings, the mechanistic conclusions taken throughout the manuscript, also placed in the final model, are not supported by the data and must be eliminated. In this respect, as detailed below, the revision is very disappointing. From my main conceptual concerns none has been correctly addressed and the responses to the technical points were only poorly answered and in some cases are even misleading.

We are puzzled by the suggestion that our revisions were considered “disappointing”, and that some questions were “poorly answered” and “in some cases are even misleading”. We took an extraordinary effort in the last revision to address all the key points, provide source data in the accompanying Excel files with 73 data sheets (in the most current version). Our last revision included a substantial amount of new data, and even more data has been added in this revision. Our intention was to maximize transparency and we definitely did not intend to mislead. We apologize if some of our conclusions are overstated and have come across as misleading. In response, we now have made additional edits and provide additional data to address the points raised by Reviewer 3.

1- Mfn2 depletion disrupts AJ junctions. As stated before, this could depend on the mitochondrial morphology defect specifically caused by Mfn2 depletion. The request to analyse EC barrier integrity upon depletion of Mfn1, Opa1, Drp1 and both fusion and fission components was not addressed.

In response, we examined whether mitochondrial morphological changes that may be the consequence of modulating mitochondrial fission or fusion affects EC barrier integrity. As suggested by the reviewer, we depleted key mitochondrial fission or fusion proteins, Drp1, Mfn1, Mfn2, and Opa1 with single, double, or triple depletion. The EC barrier integrity was evaluated by immunostaining for VE-cadherin. As shown at **Supplementary Figures 3K and L**, Mfn2 depleted

ECs clearly show disrupted VE-cadherin barrier integrity. However, depletion of Mfn1, Drp1, or Opa1 did not disrupt the barrier.

We have revised manuscript at Result section as following; “Next, we examined whether changes in the mitochondrial morphology could affect EC barrier integrity, as Mfn2 is a key regulator of mitochondrial fusion and therefore any effects on EC barrier integrity may be mediated by changing mitochondrial dynamics. We used siRNA to specifically deplete the mitochondrial fission mediator Drp1 or the mitochondrial fusion mediators Opa1, Mfn1 and Mfn2 (Supplementary Figures 3K and L). Even though the siRNA depletions were sufficient to modify mitochondrial network structure, as seen in the increase of mitochondrial fragmentation following Opa1 depletion or the increase in mitochondrial elongation with Drp1 depletion, EC barrier integrity was only significantly decreased with Mfn2 depletion suggesting that the observed Mfn2 effects on EC barrier integrity were not primarily related to the Mfn2 role in mitochondrial fusion.”

2- A fraction of Mfn2 does not co-localize with Tom20. As stated before, absence of Mfn2 co-staining with Tom20 does not exclude Mfn2 from mitochondria and does not exclude the presence of mitochondria in close proximity to adherens junctions. The fact that Mfn1 does not locate to AJ junctions is not a valid argument to exclude Mfn2 from the mitochondrial membrane. The request to analyse other mitochondrial markers (OM, IN, IMS, matrix) and to perform biochemical fractionation of mitochondria was not addressed.

In response, we examined the localization of additional mitochondrial proteins such as Drp1, VDAC, Opa1, and COXIV in resting ECs to address whether mitochondria bind to VE-cadherin at the plasma membrane. Drp1, VDAC, Opa1, and COXIV did not co-localize with VE-cadherin (Reviewer Figure R2). These data suggest that mitochondria do not co-localize with the adherens junction protein VE-cadherin but that Mfn2 behaves differently from other mitochondrial proteins. The Mfn2 colocalization with VE-cadherin supports the notion that Mfn2 is present at adherens junctions while not being in mitochondria. Next, we examine whether mitochondrial proteins change their mitochondrial localization during inflammatory condition. The ECs were stimulated with TNF α for 6 h and subcellularly fractionated to cytosol and mitochondria using mitochondrial fractionation kit (Thermo #89874). Interestingly, the amount of Mfn2 and Mfn1 at mitochondria did not change following TNF stimulation but Drp1, Opa1, and Tom20 slightly increased in mitochondria by TNF stimulation (Reviewer Figure R3). As only a small fraction of

Mfn2 is found at the plasma membrane and the bulk of Mfn2 is found in the mitochondria, it is not surprising that mitochondrial fractionation shows stable mitochondrial levels during baseline and after inflammatory stimulation.

3- The nuclear localization of full length Mfn2 (visible from their western blots) is not convincingly demonstrated. As state before, Mfn2 is an integral membrane protein. It will not “simply” translocate to the nucleus (upon disruption of the nuclear barrier, Mfn2 could e.g. be close to DNA in the cytoplasm, be inserted in vesicles present in the nucleus, ...). However, the biochemical fractionation into soluble vs membrane proteins was not addressed.

In addition to mitochondrial fractionation assay proposed by the reviewer, we also performed another subcellular fractionation assay to assess Mfn2 presence in nuclear fractions. As we showed nuclear Mfn2 under inflammatory condition in **Fig. 7** of the main manuscript using a biochemical and imaging assays, we confirmed this finding using an additional biochemical fractionation assay as suggested by the reviewer (Thermo#78840 fraction kit) (**Reviewer Figure R4**). Importantly, based on Reviewer 3’s suggestions, we also examined whether other mitochondrial proteins such as VDAC or COXIV also translocate into the nucleus. As shown in **Reviewer Figure R4**, Mfn2 clearly increased nuclear translocation following TNF stimulation, but other mitochondrial proteins such as Drp1, VDAC (also a mitochondrial membrane protein), or COXIV did not translocate into the nucleus. This indicates some degree of selectivity for Mfn2 to enter the nucleus. However, we also agree with the reviewer that we do not know the precise mechanisms of Mfn2 nuclear entry. We have revised the manuscript Discussion section as follows to acknowledge this limitation because this needs to be addressed in future studies on nuclear entry mechanisms: *“However, there are important questions regarding the mechanisms of Mfn2 nuclear translocation during inflammation that still need to be addressed. Recent studies suggest that several mitochondrial enzymes can translocate into the nucleus as a form of mitochondria-to-nucleus communication² and that mitochondria-derived vesicles or chaperone proteins and*

nuclear transcription factors may promote the entry of selected mitochondrial proteins into the nucleus via the nuclear pores^{3,4}. It is possible that the Mfn2 interaction with β -catenin or other proteins may facilitate the entry via nuclear pores during inflammation but this will need to be addressed in future studies targeting nuclear transport mechanisms”.

4- A disulphide bond between Mfn2 and beta-catenin is not demonstrated. Showing that cysteines from both Mfn2 and beta-catenin are available for sulfenylation and that their physical interaction increases upon H₂O₂ does not prove it. Beta-catenin size clearly shifts up upon oxidation, in non-reducing gels, but whether this depends on Mfn2 was not analysed. Importantly, and in contrast to beta-catenin, there is no convincing shift observed for Mfn2 in Sup fig 5. It rather suggests that oxidation leads to beta-catenin S-S formation(s) (this could easily be tested), which increases its binding to Mfn2, perhaps by exposing a bigger/better interaction interface. The request to identify the responsible cysteines was not addressed (although mass-spec analysis of their DCP-bio experiments should allow it).

We respectfully disagree with the reviewer’s comment “A disulphide bond between Mfn2 and beta-catenin is not demonstrated”. We showed that the interaction of Mfn2 and β -catenin was increased by H₂O₂ treatment and was restored by N-acetyl cysteine (NAC) which effectively reduces disulfide bonds⁵, as can be seen **Figure 5C and D**. In addition, we showed that the cysteine sulfenylation of Mfn2 and β -catenin increased by TNF α stimulation in **Figure 5E and F**. These data indicate that the interaction of Mfn2 and β -catenin during inflammation is associated with disulfide bond formation. However, in response to the reviewer’s request for additional experiments using the DCP-Bio1 assay, we further addressed the reviewer’s comments on the role of sulfenylation. Control and Mfn2 depleted (Mfn2-KD) ECs were stimulated with 500 μ M H₂O₂ for 30 min and subjected for DCP-Bio1 assay to determine cysteine sulfenylation of Mfn2 and β -catenin. H₂O₂ treated control ECs clearly induced cysteine sulfenylation of Mfn2 and β -catenin, but Mfn2 depleted ECs showed inhibition of β -catenin sulfenylation(**Reviewer Figure R5**).

Reviewer Figure R4. Effect of inflammation on Mfn2 localization. The confluence ECs were treated with TNF α for 6 h and subjected for subcellular fractionation (cytosol, membrane: plasma membrane+mitochondrial membrane, soluble nucleus extract). The equal amount of protein for subcellular fractions was loaded in SDS-PAGE to evaluate their localization. The mitochondrial proteins (Mfn2, Drp1, VDAC, and COXIV), cytosolic proteins (GAPDH), plasma membrane (Na/K ATPase) and nuclear matrix protein (p84) were determined by Western blotting with specific antibodies. memb: membrane; NE: soluble nucleus extract (NE).

Reviewer Figure R5. Mfn2 sulfenylation requires for β -catenin sulfenylation by oxidative stressed ECs. Control and Mfn2 depleted (Mfn2-KD) ECs were treated with 500 μ M H₂O₂ for 30 min and subjected to the DCP-Bio1 assay. The total sulfenylated proteins were pulled down with streptavidin agarose beads and the sulfenylation of Mfn2 and β -catenin was determined by Western blotting.

Furthermore, we examined whether the interaction of Mfn2 and β -catenin is dependent on Mfn2 under oxidative stimulation. Control and Mfn2 depleted (Mfn2-KD) ECs were stimulated with 500 μ M H_2O_2 for 30 min. Total Mfn2 was immunoprecipitated with a specific antibody and the complex of Mfn2 and β -catenin was assessed with specific antibodies for β -catenin or Mfn2 under non-reducing SDS-PAGE without β -mercaptoethanol. As shown in **New Supplementary Figure 5A**, the oxidized complexes between Mfn2 and β -catenin were increased by H_2O_2 treatment in a non-reducing gel but the complex formation was decreased by Mfn2 depletion. We also confirmed that the H_2O_2 induced interaction of Mfn2 and β -catenin is dependent on Mfn2 in reducing conditions in the **new Supplementary Figure 5B**. We replaced the old Supplementary Figures 5A and B with new data and removed old Supplementary Figures 5C and D, and revised the Figure orders accordingly. We agree that a mass spec analysis would be helpful to identify specific cysteine residues and have acknowledged this in the revised Discussion.

We have revised manuscript at Result section as follows: *“Moreover, we found that complex of Mfn2 and β -catenin showed bands shift dependent on Mfn2 in non-reducing SDS-PAGE.” “We then investigated whether the complex formation of Mfn2 and β -catenin was associated with cysteine sulfenylation, a key initial step for cysteine oxidation. To examine whether TNF α induced-ROS modulate cysteine sulfenylation of Mfn2 and β -catenin, TNF α stimulated-ECs were lysed with lysis buffer containing DCP-Bio1”.*

Further, from the previously raised technical and clarity concerns, just to mention a few:

1- I found no new review Mfn2 references, like for example PMID 32304672, 31156466, 31252211. The findings ascribed in references 13,14,15 are not relevant for this manuscript. Mfn2 clearly regulates the proximity between mitochondrial and the ER. However, regarding the localization of Mfn2 at the ER, if mentioned, it must be stated that it is controversial. Also, while

being both extremely relevant, refs 18,19 are not the most appropriate for the localization statement.

In response, we have extensively revised the Introduction to cite the Mfn2 references demanded by Reviewer 3.

2- The raw data was not provided. Just showing a selected name list does not replace for the requested identity and quantification of the proteins found.

We found all identified proteins in **Sup Fig 1C** without any selection during the last revision as we felt that this would be sufficient for the readers and reviewers. However, as requested by the reviewer, we now also include the accession number, molecular weight, total spectra counts, number of peptides, Crapome results and averaged fold change (FC) and P-Value for all 3 replicates in the **Source data Excel file for supplementary information**.

Identified Proteins	Accession Number	Alternate ID	Molecular Weight	Total Spectral Counts						# peptides						Crapome		Averaged FC and P-Value for					
				GFPAl one-Rep1	GFPAl one-Rep2	GFPAl one-Rep3	GFPX Rep1	GFPX Rep2	GFPX Rep3	GFPAl one-Rep1	GFPAl one-Rep2	GFPAl one-Rep3	GFPX Rep1	GFPX Rep2	GFPX Rep3	GFP-X/ GFP P Alone	GFP-X/ GFP P Alone	Frequency	Ave SC	AVERAGED GFP-X/ GFP P Alone	GFP-X/ GFP P Alone TTEST		
Cluster of Keratin, type I cytoskeletal 18 OS: Homo sapiens GN:KRT18 PE:1SVv2 (K18)	K1C18_HUMAN	KRT18	48 kDa	15	19	8	1	4	4	11	10	6	1	4	4	0.0887	0.205	0.5	57.9%	123	0.26	0.03	
DNA-dependent protein kinase catalytic subunit OS: Homo sapiens GN:PRKDC PE:1SV	PRKDC_HUMAN	PRKDC	469 kDa	7	7	5	11	11	8	7	7	5	11	11	8	1574	1574	16	42.56%	22	158	0.04	
Alpha-enolase OS: Homo sapiens GN:ENO1 PE:1SVv2	ENO1_HUMAN	ENO1	47 kDa	10	7	9	5	4	2	7	5	5	2	2	0.6	0.574	0.222	56.26%	177	0.46	0.03		
40S ribosomal protein S3 OS: Homo sapiens GN:RPS3A PE:1SVv2	RPS3A_HUMAN	RPS3A	30 kDa	2	4	2	7	6	4	2	4	2	7	6	4	3.5	15	2	49.39%	9	2.33	0.05	
Interleukin enhancer-binding factor 2 OS: Homo sapiens GN:ILF2 PE:1SVv2	ILF2_HUMAN	ILF2	43 kDa	1	4	4	7	7	5	1	3	4	7	6	4	7	175	128	36.86%	55	3.33	0.05	
Phenylalanine-tyrosine ligase alpha subunit OS: Homo sapiens GN:FAHSA PE:1SVv3	TYFA_HUMAN	FAHSA	55 kDa	2	2	1	4	3	3	2	2	1	4	3	3	2	15	3	16.66%	23	2.17	0.03	
Sulfide-quinone oxidoreductase, mitochondrial OS: Homo sapiens GN:SQORL PE:1SVv1	SQORL_HUMAN	SQORL	50 kDa	1	2	1	3	3	2	1	2	1	3	3	2	3	15	2	4.45%	5	2.17	0.05	
Cluster of Histone H2A type 1-B/E OS: Homo sapiens GN:HIST1H2AB PE:1SVv2 (H2AB)	H2AB_HUMAN	HIST1H2AB	14 kDa	2	2	2	3	4	3	1	1	1	2	2	2	15	2	15.854%	10	1.67	0.02		
Serpin OS: Homo sapiens GN:SERP1 PE:1SVv2	SERP1_HUMAN	Serpin-5	55 kDa	1	1	1	5	3	3	1	1	1	5	3	2	5	3	2	19.22%	47	2.33	0.06	
Microtubule-actin cross-linking factor 1 isoforms 1/2/3 OS: Homo sapiens GN:MACF1	MACF1_HUMAN	MACF1	858 kDa	1	1	2	5	6	2	1	1	2	5	6	2	5	5	3.45%	49	3.67	0.05		
60S ribosomal protein L27a OS: Homo sapiens GN:RPL27A PE:1SVv2	RPL27A_HUMAN	RPL27A	17 kDa	0	1	0	2	2	1	0	1	0	2	2	1	#DIV/0!	#DIV/0!	40.16%	32	#DIV/0!	0.05		
Cytochrome-associated protein 4 OS: Homo sapiens GN:CA4 PE:1SVv2	CA4_HUMAN	CA4	66 kDa	4	1	5	5	0	7	4	1	5	5	0	7	125	9	14	16.19%	5	3.55	0.08	
Protein PML OS: Homo sapiens GN:PML PE:1SVv3	PML_HUMAN	PML	58 kDa	0	0	0	3	1	1	0	0	0	2	1	#DIV/0!	#DIV/0!	5.15%	12	#DIV/0!	0.02			
Digitinin OS: Homo sapiens GN:DYSF PE:1SVv1	DYSF_HUMAN	DYSF	237 kDa	0	0	0	3	1	2	0	0	0	3	1	2	#DIV/0!	#DIV/0!	0.72%	1	#DIV/0!	0.03		
BAG family molecular chaperone regulator 2 OS: Homo sapiens GN:BAG2 PE:1SVv1	BAG2_HUMAN	BAG2	24 kDa	0	0	0	3	3	1	1	0	0	3	3	1	2	#DIV/0!	#DIV/0!	23.36%	23	#DIV/0!	0.05	
Microuran OS: Homo sapiens GN:URAN PE:1SVv3	URAN_HUMAN	URAN	85 kDa	0	0	0	68	52	24	0	0	0	28	26	15	#DIV/0!	#DIV/0!	0.24%	17	#DIV/0!	0.02		
E3 ubiquitin-protein ligase MARCKS OS: Homo sapiens GN:MARCKS PE:1SVv1	MARCKS_HUMAN	Marck-5	31 kDa	0	0	0	5	4	3	0	0	0	5	4	2	#DIV/0!	#DIV/0!	#DIV/0!	#DIV/0!	#DIV/0!	0.05		
Coactivator of transcription factor 1 OS: Homo sapiens GN:CFAM20	CFAM20_HUMAN	CFAM20	122 kDa	1	0	1	2	2	3	1	0	1	3	2	3	2	#DIV/0!	#DIV/0!	3	0.52%	61	#DIV/0!	0.05
60S ribosomal protein L32 OS: Homo sapiens GN:RPL32 PE:1SVv2	RPL32_HUMAN	RPL32	15 kDa	0	0	1	2	2	1	0	0	1	2	3	1	#DIV/0!	#DIV/0!	1.00	0.175%	51	#DIV/0!	0.03	
Insulin-like growth factor 2 mRNA-binding protein 3 OS: Homo sapiens GN:IGFBP3 PE:1SVv1	IGFBP3_HUMAN	IGFBP3	64 kDa	0	0	2	2	3	2	0	0	2	2	3	2	#DIV/0!	#DIV/0!	1.00	0.217	45	#DIV/0!	0.05	
Ple-mRNA-processing factor 1 OS: Homo sapiens GN:PPF1 PE:1SVv1	PPF1_HUMAN	PPF1	55 kDa	0	0	1	2	3	1	0	0	1	2	3	1	#DIV/0!	#DIV/0!	1.00	0.294	7	#DIV/0!	0.03	
Regulator of nonsense transcripts 1 OS: Homo sapiens GN:RNT1 PE:1SVv2	RNT1_HUMAN	RNT1	12 kDa	0	0	0	2	1	1	0	0	0	2	1	1	#DIV/0!	#DIV/0!	0.0852	51	#DIV/0!	0.02		
Signal recognition particle receptor subunit beta OS: Homo sapiens GN:SRPB PE:1SV	SRPB_HUMAN	SRPB	30 kDa	0	0	0	1	2	1	0	0	0	1	2	1	#DIV/0!	#DIV/0!	0.0852	17	#DIV/0!	0.02		
Dolichol phosphate mannosyltransferase subunit 1 OS: Homo sapiens GN:DPMT1 PE:1SV	DPMT1_HUMAN	DPMT1	30 kDa	0	0	0	1	1	1	0	0	0	1	1	1	#DIV/0!	#DIV/0!	0.0631	13	#DIV/0!	#DIV/0!		
Lamin-B2 OS: Homo sapiens GN:LMNB2 PE:1SVv4	LMNB2_HUMAN	LMNB2	70 kDa	0	0	0	1	2	1	0	0	0	1	2	1	#DIV/0!	#DIV/0!	0.1655	44	#DIV/0!	0.02		

*Analyzed each by LC-MS/MS.
 *Processed data using Proteome Discoverer 2.1 - searched data against a Uniprot reviewed human database.
 *Imported data into Scaffold 4. Spectral counts were calculated and used for relative quantitation.
 *Results were further analyzed in Excel.
 *All proteins were searched against the Crapome (crapome.org), a common repository for affinity purification mass spec contaminants. A crapome frequency was calculated - the lower the frequency, the more likely the protein is a true hit.
 *Calculated the fold change (FC) for each replicate using absolute spectral counts: GFP-X/GFP alone.
 *Column V includes the averaged fold change.
 *Also calculated p-value using t-test, results in column W (two-tailed, homoscedastic)
 *Using p-value cut-off of 0.1, the following tabs contain higher confidence hits (based on additional stringency of p-value, less proteins reported than before)
 GFP-X: Hit proteins that had FC > 1.5 of FC < 0.6 for GFP-X/GFP alone, #DIV/0! means the protein was not identified in the GFP-alone sample. Must meet this criteria for at least 2 out of 3 replicates.
 *25 proteins.

Raw data for proteomic analysis. The table includes protein identification name accession number, molecular weight, total spectra counts, number of peptides, Crapome results and averaged fold change (FC) and P-Value. GFP is control and GFP-X means GFP-Mfn2.

3- Regarding GTPase activity, it was not convincing before. What the authors call new sup fig 5C and D was already there before as sup fig. 3 B and C. Thus, showing exactly the same panels, does not help and is inappropriate.

As we described in the last revision, the GTPase assay is a well established method^{2, 7, 8}. We purified endogenous Mfn2 by immunoprecipitation assay after TNF α stimulation and the pulled-down Mfn2 was used to measure GTPase activity. We also used a negative control which used control IgG nonspecific effects. We found that Mfn2 GTPase activity decreases following TNF α stimulation in a time dependent manner and recovered at 24 h. Thus, the GTPase assay specifically demonstrates the pulled-down Mfn2 enzyme activity. The data was used for initial submission and renamed them as **new Supplemental Figures Fig 5C and 5D** to indicate the new figure numbering as part of the reorganization of the data. We do not understand why such reorganization would be "inappropriate" but in deference to the reviewer's demands, we have now

removed the GTPase data from Sup Fig 5C and D and the manuscript. We now discuss this issue in the Discussion section because Mfn2 sulfenylation during inflammation could impact Mfn2 GTPase activity and could be of interest to Mfn2 researchers studying the GTPase activity. In addition to identifying the relevant cysteine residues, future studies building on our work could also how Mfn2 sulfenylation affects its GTPase activity.

We have revised Discussion section as following: “*We found that TNF α -induced ROS increased Mfn2 sulfenylation in ECs. Our results suggest that the sulfenylation step might be required for the disassociation of Mfn2 from adherens junctions and that Mfn2 sulfenylation may be corelated with its GTPase activity which may reflect new post-translational regulation during inflammation. Thus, there are warranted to identify the responsible cysteine residues of Mfn2 to oxidative stress and to investigate how Mfn2 sulfenylation affects to its GTPase activity and mitochondrial dynamics in pathophysiological condition in future study. In addition, it is worth to address this possibility using Mfn2 GTPase dominant negative mutant (DN) even though it will be very challenge because Mfn2 GTPase-DN can inhibit mitochondrial fusion*”.

Finally, some of the new data presented lacks controls and is overstated.

In the previous revision, we used controls for all key experiments (siRNA versus scramble siRNA, control IgG, LoxP mice versus Cre/LoxP mice, etc). In the current revision, we have tried to accede to all the demands of Reviewer 3, including the removal of data such as the GTPase data and we have accentuated limitations and the need for additional experiments that lie beyond the scope of the current manuscript. We also removed the model figure from the manuscript which was criticized by Reviewer 3 as overstating some of the mechanisms. We hope that this revision now addressed these concerns.

References

1. Ghasemi, A. & Zahediasl, S. Normality tests for statistical analysis: a guide for non-statisticians. *Int J Endocrinol Metab* **10**, 486-489 (2012).
2. Kim, K.H., Son, J.M., Benayoun, B.A. & Lee, C. The Mitochondrial-Encoded Peptide MOTS-c Translocates to the Nucleus to Regulate Nuclear Gene Expression in Response to Metabolic Stress. *Cell Metab* **28**, 516-524 e517 (2018).
3. Nagaraj, R. *et al.* Nuclear Localization of Mitochondrial TCA Cycle Enzymes as a Critical Step in Mammalian Zygotic Genome Activation. *Cell* **168**, 210-223 e211 (2017).
4. Cardamone, M.D. *et al.* Mitochondrial Retrograde Signaling in Mammals Is Mediated by the Transcriptional Cofactor GPS2 via Direct Mitochondria-to-Nucleus Translocation. *Mol Cell* **69**, 757-772 e757 (2018).
5. Tersteeg, C. *et al.* N-acetylcysteine in preclinical mouse and baboon models of thrombotic thrombocytopenic purpura. *Blood* **129**, 1030-1038 (2017).
6. Mattie, S., Krols, M. & McBride, H.M. The enigma of an interconnected mitochondrial reticulum: new insights into mitochondrial fusion. *Curr Opin Cell Biol* **59**, 159-166 (2019).
7. Borg Distefano, M. *et al.* TBC1D5 controls the GTPase cycle of Rab7b. *J Cell Sci* **131** (2018).
8. Duan, C. *et al.* Mdivi-1 attenuates oxidative stress and exerts vascular protection in ischemic/hypoxic injury by a mechanism independent of Drp1 GTPase activity. *Redox Biol* **37**, 101706 (2020).

Reviewers' comments:

Reviewer #2 (Remarks to the Author):

The authors have responded to my concerns, in a satisfactory manner.

Reviewer #3 (Remarks to the Author):

The present revised version is substantially improved, concerning the specific role of MFN2 in EC barrier integrity and protection from inflammation. I would even suggest directly referring in the text to the fact that even by restoring mt morphology (by Drp1 plus Mfn2 simultaneous KD) EC integrity is not rescued. Importantly, however, these conclusions also depend on figures R4 and R5. Thus, rather than only available to the reviewer, it is essential to insert these data in the manuscript. In addition, the two excel files should also be added (presently they are not referred, neither in the main manuscript, in the mat met or in the sup file).

Importantly, the overstatements have been nearly eliminated. However, even if beta-caterin sulfenylation depends on Mfn2, a direct interaction of Mfn2 with beta-caterin via S-S bonds is still not proven. I would suggest deeming their statements in this regard. Finally “Mfn2 translocation to the nucleus” was not shown. Alternatively, the authors could state “Mfn2 accumulates in the nuclear fraction”. This includes revising also the abstract. Indeed, it is not demonstrated that the fraction of Mfn2 present at the mt, AJ or Nucleus has ever been in the other locations. Newly translated Mfn2 could directly account for the different locations.

Minor point:

To convincingly show the difference between Mfn2 and other mt proteins, please add a shorter exposure of Mfn2 to figure “R4”, with an exposure in the membrane fraction similar to the one of VDAC and Cox4.

Reviewer #3 (Remarks to the Author):

The present revised version is substantially improved, concerning the specific role of MFN2 in EC barrier integrity and protection from inflammation. I would even suggest directly referring in the text to the fact that even by restoring mt morphology (by Drp1 plus Mfn2 simultaneous KD) EC integrity is not rescued. Importantly, however, these conclusions also depend on figures R4 and R5. Thus, rather than only available to the reviewer, it is essential to insert these data in the manuscript. In addition, the two excel files should also be added (presently they are not referred, neither in the main manuscript, in the mat met or in the sup file). Importantly, the overstatements have been nearly eliminated. However, even if beta-caterin sulfenylation depends on Mfn2, a direct interaction of Mfn2 with beta-caterin via S-S bonds is still not proven. I would suggest deeming their statements in this regard. Finally “Mfn2 translocation to the nucleus” was not shown. Alternatively, the authors could state “Mfn2 accumulates in the nuclear fraction”. This includes revising also the abstract. Indeed, it is not demonstrated that the fraction of Mfn2 present at the mt, AJ or Nucleus has ever been in the other locations. Newly translated Mfn2 could directly account for the different locations.

Minor point:

To convincingly show the difference between Mfn2 and other mt proteins, please add a shorter exposure of Mfn2 to figure “R4”, with an exposure in the membrane fraction similar to the one of VDAC and Cox4.

Point-by-point responses:

1. Importantly, however, these conclusions also depend on figures R4 and R5. Thus, rather than only available to the reviewer, it is essential to insert these data in the manuscript.

We agree with reviewer's comment and have inserted these data in Supplementary figure 6c for figure R4 and Supplementary figure 5c for figure R5.

2. In addition, the two excel files should also be added (presently they are not referred, neither in the main manuscript, in the mat met or in the sup file).

The Source Data now includes two excel files for raw data, two PPT files for full microscopy images, and one supplementary Table 1 listing all primers.

3. Importantly, the overstatements have been nearly eliminated. However, even if beta-caterin sulfenylation depends on Mfn2, a direct interaction of Mfn2 with beta-caterin via S-S bonds is still not proven. I would suggest deeming their statements in this regard.

We revised our statements about disulfide bond formation between Mfn2 and beta-catenin in first paragraph of page 10:

“...ROS mediate the interaction of Mfn2 and β -catenin.” and “...Mfn2 depletion of ECs resulted in the inhibition of β -catenin sulfenylation.”

4. Finally “Mfn2 translocation to the nucleus” was not shown. Alternatively, the authors could state “Mfn2 accumulates in the nuclear fraction”. This includes revising also the abstract. Indeed, it is not demonstrated that the fraction of Mfn2 present at the mt, AJ or Nucleus has ever been in the other locations. Newly translated Mfn2 could directly account for the different locations.

We changed all “translocation” terms to “accumulation” in the Manuscript.

5. Minor point: To convincingly show the difference between Mfn2 and other mt proteins, please add a shorter exposure of Mfn2 to figure “R4”, with an exposure in the membrane fraction similar to the one of VDAC and Cox4.

We replaced the Mfn2 blot with a new blot that has shorter exposure time in Supplementary Figure 6c.